# The Hippo pathway controls myofibril assembly and muscle fiber growth by regulating sarcomeric gene expression

Aynur Kaya-Çopur[1,2]*, Fabio Marchiano[1], Marco Y Hein[2], Daniel Alpern[3], Julie Russeil[3], Nuno Miguel Luis[1], Matthias Mann[2], Bart Deplancke[3], Bianca H Habermann[1], Frank Schnorrer[1,2]*

[1]Aix Marseille University, CNRS, IBDM, Turing Center for Living Systems, Marseille, France; [2]Max Planck Institute of Biochemistry, Martinsried, Germany; [3]Institute of Bioengineering, School of Life Sciences, École Polytechnique Fédérale de Lausanne (EPFL), Lausanne, Switzerland

**\*For correspondence:**
aynur.KAYA-COPUR@univ-amu.fr (AK-Ç);
frank.schnorrer@univ-amu.fr (FS)

**Competing interests:** The authors declare that no competing interests exist.

**Abstract** Skeletal muscles are composed of gigantic cells called muscle fibers, packed with force-producing myofibrils. During development, the size of individual muscle fibers must dramatically enlarge to match with skeletal growth. How muscle growth is coordinated with growth of the contractile apparatus is not understood. Here, we use the large *Drosophila* flight muscles to mechanistically decipher how muscle fiber growth is controlled. We find that regulated activity of core members of the Hippo pathway is required to support flight muscle growth. Interestingly, we identify Dlg5 and Slmap as regulators of the STRIPAK phosphatase, which negatively regulates Hippo to enable post-mitotic muscle growth. Mechanistically, we show that the Hippo pathway controls timing and levels of sarcomeric gene expression during development and thus regulates the key components that physically mediate muscle growth. Since Dlg5, STRIPAK and the Hippo pathway are conserved a similar mechanism may contribute to muscle or cardiomyocyte growth in humans.

## Introduction

Mammalian skeletal muscles are built from gigantic cells called muscle fibers, up to several centimetres long, that mechanically link distant skeletal elements. Muscle forces are produced by highly regular molecular arrays of actin, myosin and titin filaments called sarcomeres. Each sarcomere has a length of about 3 µm in relaxed human skeletal muscles (*Ehler and Gautel, 2008*; *Llewellyn et al., 2008*; *Regev et al., 2011*). Thus, hundreds of sarcomeres need to assemble into long chains called myofibrils in order to generate force across the entire muscle fiber (*Lemke and Schnorrer, 2017*). Large muscle fibers contain many parallel myofibrils, which are laterally aligned to a cross-striated pattern to effectively power animal locomotion (*Gautel, 2008*; *Schiaffino et al., 2013*). How muscle fibers grow to these enormous sizes and how their growth is coordinated with the assembly and maturation of the individual myofibrils within the muscle is a challenging biological problem that is not well understood.

Muscle fibers are built during animal development. Initially, many small myoblasts fuse to myotubes, whose long-ends then mechanically connect to tendon cells (*Kim et al., 2015*; *Schnorrer and Dickson, 2004*). This enables the build-up of mechanical tension within myotubes, which consecutively triggers myofibril assembly and the transition of myotubes to early myofibers (*Weitkunat et al., 2017*; *Weitkunat et al., 2014*). Following myofibril assembly, the immature myofibrils mature and build functional sarcomeres. To do so, each myofibril grows in length and diameter and thereby supports the extensive muscle fiber growth during embryonic and postembryonic

development (*González-Morales et al., 2019*; *Orfanos et al., 2015*; *Reedy and Beall, 1993*; *Sanger et al., 2017*; *Sparrow and Schöck, 2009*). For the correct developmental sequence of myofibril morphogenesis, the protein concentrations of the various sarcomeric components need to be precisely regulated (*Orfanos and Sparrow, 2013*; *Schönbauer et al., 2011*). This is particularly prominent in mammalian muscle fibers, in which sarcomeric proteins transcriptionally switch isoforms from embryonic to neonatal and finally to adult isoforms (*Schiaffino, 2018*; *Schiaffino et al., 2015*). In *Drosophila* indirect flight muscles, transcription of sarcomeric and mitochondrial protein coding genes starts just before myofibril assembly and is then strongly boosted during myofibril maturation, when myofibrils grow in length and width (*González-Morales et al., 2019*; *Shwartz et al., 2016*; *Spletter et al., 2018*). Concomitantly with the growth of the myofibrils, the T-tubule network forms (*Peterson and Krasnow, 2015*; *Sauerwald et al., 2019*) and also the mitochondria grow in size (*Avellaneda et al., 2020*; *Spletter et al., 2018*). How this precise transcriptional control is achieved and coordinated with muscle fiber growth is unclear.

One central pathway controlling organ size during development and tumorigenesis is the Hippo pathway, which regulates the activity of the growth promoting transcriptional co-activator Yorkie (Yki, YAP and TAZ in mammals) (*Pan, 2010*; *Zanconato et al., 2019*). The core of the pathway is composed of a kinase cascade with Hippo (Hpo; Mst1 and Mst2 in mammals) phosphorylating the downstream kinase Warts (Wts; Lats1 and Lats2 in mammals) (*Udan et al., 2003*; *Wu et al., 2003*). Phosphorylated Wts is active and in turn phosphorylates Yki (*Huang et al., 2005*), leading to the cytoplasmic retention of phospho-Yki by 14-3-3 proteins (*Dong et al., 2007*; *Oh and Irvine, 2008*; *Ren et al., 2010*). When the pathway is not active, unphosphorylated Yki enters into the nucleus, binds to the Tead protein Scalloped (Sd), and turns on transcriptional targets (*Goulev et al., 2008*; *Wu et al., 2008*; *Zhang et al., 2008*). The majority of these targets promote organ growth by suppressing apoptosis and stimulating cell growth and cell proliferation (*Harvey and Tapon, 2007*).

A key control step of the Hippo pathway is the localisation and kinase activity of Hippo. In epithelial cells, the scaffold protein Salvador promotes Hippo kinase activity by localising Hippo to the plasma membrane (*Yin et al., 2013*) and by inhibiting a large protein complex called the STRIPAK (Striatin-interacting phosphatase and kinase) complex (*Bae et al., 2017*). The STRIPAK complex contains PP2A as active phosphatase, which dephosphorylates a key Hippo auto-phosphorylation site and thus inhibits Hippo activity (*Ribeiro et al., 2010*; *Zheng et al., 2017*). dRassf can promote the recruitment of STRIPAK to Hippo and thus inactivate Hippo (*Polesello et al., 2006*). Furthermore, the Hippo pathway can also be regulated downstream by membrane localisation of the kinase Warts by Merlin binding, which promotes Warts phosphorylation by Hippo and thus activation of the pathway (*Yin et al., 2013*). Finally, mechanical stretch of the epithelial cell cortex was shown to directly inhibit the Hippo pathway, likely mediated by the spectrin network at the cortex, promoting nuclear localisation of Yorkie (*Fletcher et al., 2018*; *Fletcher et al., 2015*). Despite this detailed knowledge about Hippo regulation in proliferating epithelial cells, little is known about how the Hippo pathway is regulated during post-mitotic muscle development and how it impacts muscle growth.

Here, we employ a systematic in vivo muscle-specific RNAi screen and identify various components of the Hippo pathway as essential post-mitotic regulators of flight muscle morphogenesis. We find that loss of Dlg5 or of the STRIPAK complex member Slmap, which interacts with Dlg5, as well as loss of the transcriptional regulator Yorkie results in too small muscles. These small muscles express lower levels of sarcomeric proteins and as a consequence contain fewer and defective myofibrils. Conversely, over-activation of Yorkie, either by removing the negative regulators Hippo or Warts or by enabling constitutive nuclear entry of Yorkie results in premature and excessive expression of sarcomeric proteins and consequently in chaotic myofibril assembly. Therefore, our findings suggest that the Hippo pathway contributes to the precise timing of sarcomeric gene expression and thus can generate a feedback mechanism for muscles to precisely coordinate sarcomeric protein levels during myofibril assembly and maturation. This provides an attractive mechanism for how regulated transcription can coordinate muscle growth with myofibril morphogenesis.

## Results

### Growth of *Drosophila* flight muscles

We chose the *Drosophila* indirect flight muscles to investigate post-mitotic muscle fiber growth. These muscles consist of two groups, the dorsal-longitudinal flight muscles (DLMs) and the dorso-ventral flight muscles (DVMs). Both groups form in the second thoracic segment during pupal development and despite differences during myoblast fusion and myotube attachment determining their location in the thorax, their development after 24 hr after puparium formation (APF) is very similar (*Dutta et al., 2004*; *Fernandes et al., 1991*; *Schönbauer et al., 2011*). Thus, we focused our studies on the DLMs and for simplicity call them flight muscles in the remainder of the manuscript.

In order to quantify muscle fiber growth we measured fiber length and cross-sectional area of wild-type DLM flight muscles. At 24 hr APF (at 27°C) myoblast fusion is largely finished (*Weitkunat et al., 2014*), and the fibers have a length of about 270 µm and a cross-sectional area of about 1000 µm$^2$ (*Figure 1A,C*, *Supplementary file 1*). Then, flight muscles build-up mechanical tension, compact to about 220 µm in length, while their diameter grows to about 2000 µm$^2$, and assemble the immature myofibrils at 32 hr APF (*Lemke et al., 2019*; *Weitkunat et al., 2014*; *Figure 1A, C*). After 32 hr, flight muscles grow about 2.5 times in length while keeping a similar diameter until 48 hr APF. After 48 hr APF, they grow further to about 800 µm in length while increasing in diameter to almost 4000 µm$^2$ until 90 hr APF, which is shortly before eclosion at 27°C (*Figure 1A*, *Supplementary file 1*; *Spletter et al., 2018*). Thus, in total, the volume of the individual muscle fibers increases more than 10-fold in less than 3 days (*Figure 1A*, *Supplementary file 1*). Thus, indirect flight muscles are a good model to study rapid post-mitotic muscle fiber growth.

### *Dlg5* and *Slmap* are essential for flight muscle morphogenesis

To identify regulators of muscle growth, we have investigated genes identified in a genome-wide muscle-specific RNAi study that had resulted in flightless or late developmental lethality when knocked-down using muscle-specific *Mef2*-GAL4 (*Schnorrer et al., 2010*). Our analysis identified two genes, *Dlg5* (*Discs large 5*, *CG6509*) and *Slmap* (*Sarcolemma associated protein*, *CG17494*), which are conserved from *Drosophila* to human and when knocked-down using several independent RNAi constructs result in viable but completely flightless flies (*Figure 1—figure supplement 1*, *Supplementary file 1*). Inspection of the thoraces of these animals revealed complete flight muscle atrophy in pupae at 90 hr APF (*Figure 1B*). Expression of a *UAS-Dlg5-GFP* but not a *UAS-GFP-Gma* (globular moesin actin binding domain fused to GFP) control construct, was able to rescue the number of muscle fibers of *Dlg5* knock-down (*Dlg5-IR-1*) flies to wild type providing further strong evidence for the specificity of the knock-down phenotype (*Figure 1D*). We conclude that *Dlg5* and *Slmap* are two conserved genes essential for flight muscle morphogenesis during pupal stages.

To identify the developmental time point when *Dlg5* and *Slmap* are required, we analysed pupal stages and found that at 24 hr APF all flight muscles are present after *Dlg5* or *Slmap* knockdown. However, the fibers are more than 20% longer than wild type and fail to compact at 32 hr APF when myofibrils normally assemble (*Figure 1B,C*, *Supplementary file 1*). Interestingly, after 32 hr APF, when wild-type myofibers strongly grow in length, *Dlg5* and *Slmap* knock-down fibers undergo complete flight muscle atrophy until 48 hr APF (*Figure 1B*). Taken together, these data demonstrate that *Dlg5* and *Slmap* play an essential role during stages of myofibril assembly and muscle fiber growth.

### Dlg5 and Slmap partially co-localise with Dlg1 during T-tubule morphogenesis

We aimed to determine the subcellular localisation of Dlg5 and Slmap proteins in developing flight muscles and thus raised polyclonal antibodies against both proteins. Both antibodies are specific as the Dlg5 or Slmap intensities in the flight muscles, but not in motor neurons or tendons, are strongly reduced in the respective muscle-specific *Dlg5* or *Slmap* knock-down at 32 hr APF (*Figure 2—figure supplement 1A,B*). This also demonstrates the efficiency of the muscle-specific *Dlg5* or *Slmap* knock-down at protein level.

Detailed immunostainings revealed that both Dlg5 and Slmap proteins largely localise to membranous structures called T-tubules at 90 hr APF (*Figure 2A,B*), which are marked by the Dlg5-

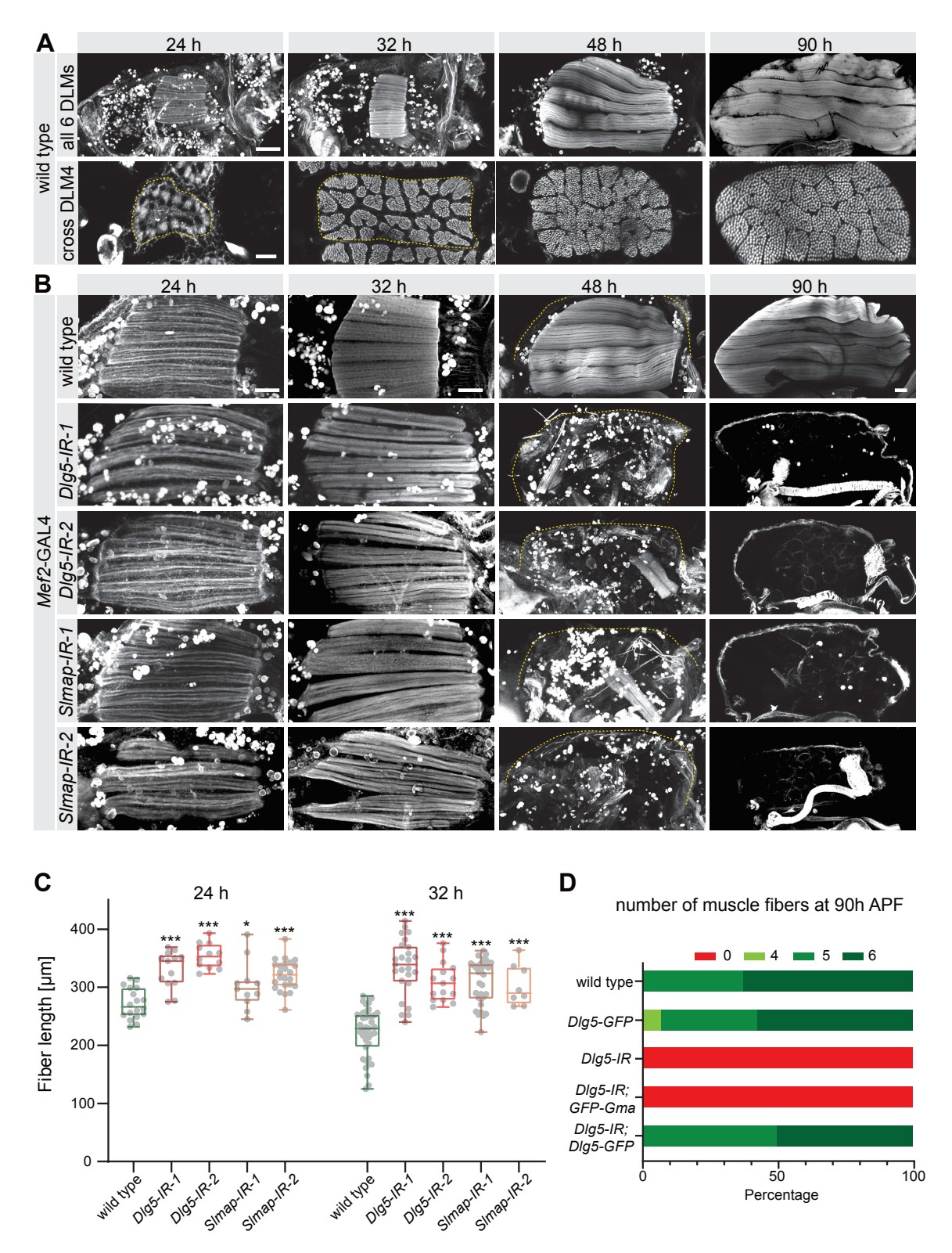

**Figure 1.** *Dlg5* and *Slmap* are essential for flight muscle morphogenesis. (**A**) Time-course of wild-type dorsal longitudinal indirect flight muscle (DLM) development. Longitudinal sections (upper panel) of all DLMs and cryo cross-sections (lower panel) of DLM4 were stained for actin. Note the muscle fiber growth in length and width. Scale bars represent 100 μm for longitudinal and 10 μm for cross-sections. (**B**) Longitudinal views of developing flight muscles at 24 hr, 32 hr, 48 hr, and 90 hr after puparium formation (APF) of wild-type, *Dlg5*, or *Slmap* knockdown genotypes (independent RNAi lines *IR-*

*Figure 1 continued on next page*

*Figure 1 continued*

1 and 2) stained for actin. Note that *Dlg5-IR* and *Slmap-IR* muscles are too long at 24 hr and 32 hr APF, and are lost after 32 hr APF. The dotted lines highlight the cuticle. Scale bars represent 50 μm. (C) Box plot showing flight muscle fiber length at 24 hr and 32 hr APF of the indicated genotypes. Each dot represents the average flight muscle length from one pupa (see Materials and methods). Box extends from 25% to 75%, line marks median, whiskers extend from max to min, all data points superimposed. Student's t-test, *** p-value<0.001, *p value < 0.05. All following box plots are plotted the same way. N ≥ 8 pupae for each genotype. (D) Number of flight muscle fibers in half thoraces at 90 hr APF of the indicated genotypes. Note that *UAS-Dlg5-GFP* but not *UAS-GFP-Gma* rescues the fiber atrophy phenotype of *Dlg5* knockdown (*Dlg1-IR-1*, in all the following figures *IR* refers to *IR-1*). The online version of this article includes the following figure supplement(s) for figure 1:

**Figure supplement 1.** Dlg5 and Slmap proteins are conserved and required for flight.

related protein Dlg1 and the membrane-binding protein Amphiphysin (*Figure 2C*; *Razzaq et al., 2001*). The mature T-tubules are in intimate contact with the mature myofibrils at their sarcomeric A-bands (*Figure 2D*; *Razzaq et al., 2001*). Interestingly, at 32 hr and 48 hr APF when first myofibrils and then T-tubules assemble, Dlg5 and Slmap proteins partially co-localise with Dlg1 (*Figure 2A,B*), however, both proteins show a more vesicular pattern compared to Dlg1 or Amphiphysin at these stages (*Figure 2C*). Together, these data show that Dlg5 and Slmap are expressed during muscle development and localise at or close to membranous structures forming the T-tubules that will closely interact with the maturing myofibrils.

## Dlg5 interacts with Slmap, a STRIPAK complex member, in *Drosophila* muscle

To define a molecular mechanism for how Dlg5 regulates muscle morphogenesis, we performed GFP immunoprecipitation followed by mass-spectrometry, using the functional *UAS-Dlg5-GFP*, which was expressed in pupal muscles with *Mef2*-GAL4. Interestingly, we not only identified Slmap as a binding partner of Dlg5 in muscle, but also Fgop2, GckIII, Striatin (Cka), and the catalytic sub-unit of PP2A phosphatase (Mts) (*Figure 3A,B*, *Supplementary file 1*). All these proteins are members of the STRIPAK complex and have been described to interact closely in mammals (*Hwang and Pallas, 2014*). This suggests that the composition of the STRIPAK complex in fly muscles is similar to mammals (*Figure 3C*), and Dlg5 is either a core member or closely interacts with this complex in muscle.

To functionally test if members of the STRIPAK complex other than Dlg5 and Slmap play a similarly important role in flight muscles, we knocked-down various component members and found that knock-down of *Striatin* (*Cka*) and *Strip* indeed result in very similar phenotypes to *Dlg5* and *Slmap* knock-down: flight muscles at 32 hr APF are longer than wild type, suggesting a fiber compaction defect, and undergo muscle atrophy after 32 hr APF (*Figure 3D,E*). Together, these data show that Dlg5 interacts with the STRIPAK complex in flight muscles, of which several members including Slmap are important for flight muscle morphogenesis.

## The Hippo pathway regulates the developmental timing of muscle morphogenesis

As the STRIPAK complex was shown to dephosphorylate and thus inactivate Hippo (*Ribeiro et al., 2010*; *Zheng et al., 2017*), we wondered if the muscle morphogenesis phenotype we observed could be linked to a function of the Hippo pathway in growing flight muscles. When knocking-down the Hippo pathway transcriptional co-activator *yorkie* in muscles, we observed a failure of muscle compaction at 32 hr APF, flight muscle atrophy at 48 hr APF and consequently flightless adults (*Figure 4A,B*, *Figure 4—figure supplement 1A*, *Supplementary file 1*). The myofiber compaction defect of *yorkie* knock-down muscles is corroborated by fiber cross-sections at 24 hr and 32 hr APF revealing much thinner muscles and phenocopying the *Dlg5* and *Slmap* loss of function (*Figure 4—figure supplement 1B*).

In contrast to the myofiber compaction defect of *yorkie* knock-down muscles, expression of an activated form of Yorkie (*yorkie-CA*), which cannot be phosphorylated by Warts, results in premature muscle fiber compaction already at 24 hr APF and strongly hyper-compacted muscle fibers at 32 hr APF, and later in disorganised myofibers (*Figure 4A,B*). The increased cross-sectional area is particularly obvious in cross-sections of *yorkie-CA* fibers (*Figure 4—figure supplement 1B*). Importantly,

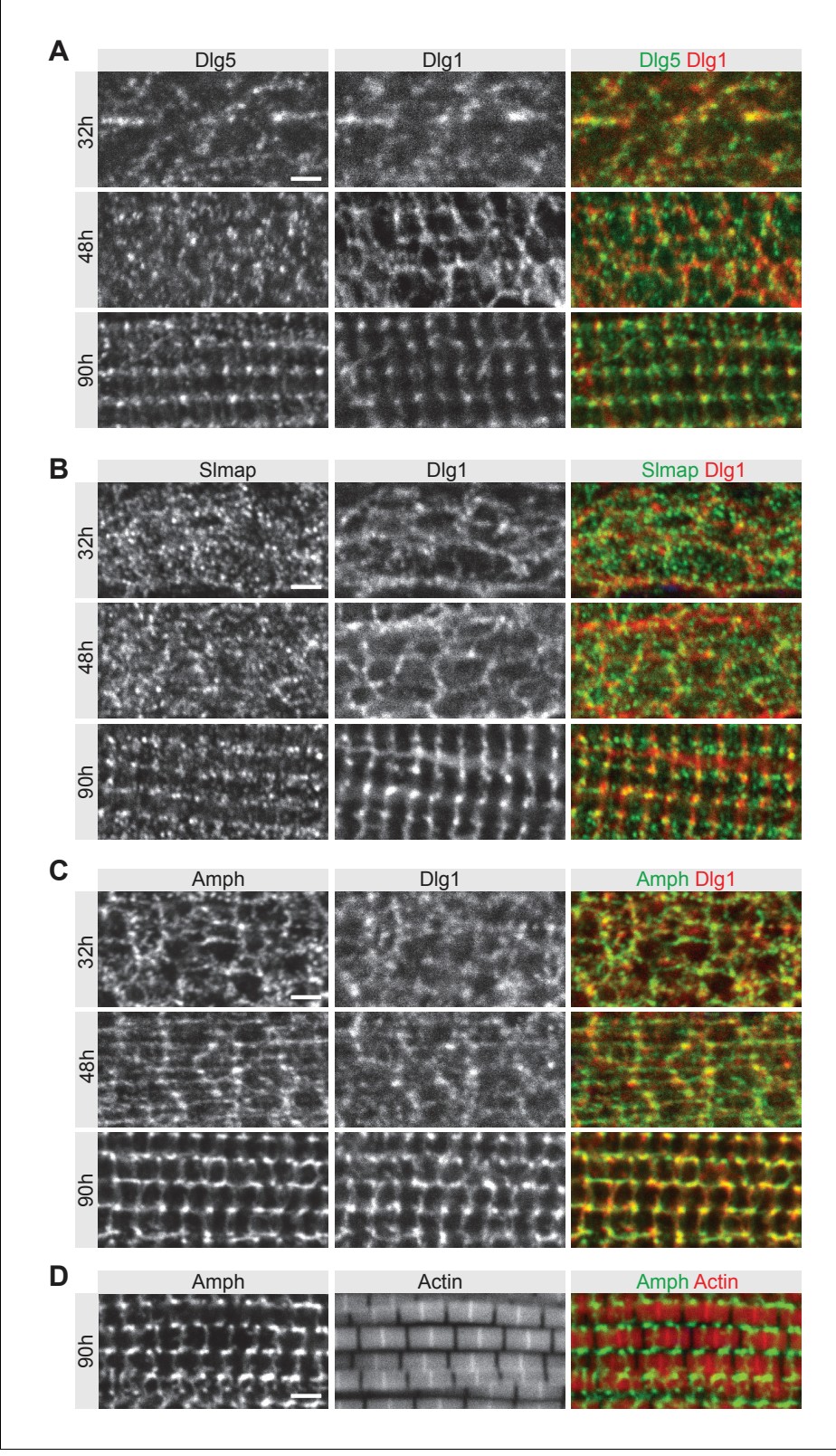

**Figure 2.** Dlg5 and Slmap partially co-localise with Dlg1 during T-tubule morphogenesis. (A–C) Wild-type developing flight muscles at 32 hr, 48 hr, and 90 hr after puparium formation (APF) co-stained for T-tubule marker Dlg1 with Dlg5 (A) or Slmap (B) or with another T-tubule marker Amphiphysin (C). (D) Wild-type flight muscles at 90 hr APF co-stained for Amphiphysin and actin with phalloidin. Scale bars represent 2 μm.

*Figure 2 continued on next page*

*Figure 2 continued*

The online version of this article includes the following figure supplement(s) for figure 2:

**Figure supplement 1.** Dlg5 and Slmap protein knock-down efficiency.

we observed the same phenotypes after knock-down of each of the two kinases *hippo* and *warts*, both negative regulators of Yorkie nuclear entry (*Figure 4A,B*). Consistently, a transcriptomics time-course of dissected developing flight muscles (*Spletter et al., 2018*) showed that *Dlg5, Slmap, hippo, warts, yorkie* and also the transcriptional activator *scalloped* are all expressed in developing flight muscles (*Figure 4—figure supplement 1C*). This strongly suggests that the Hippo pathway, by regulating phosphorylation of the transcriptional co-activator Yorkie, is essential for the correct developmental timing of flight muscle morphogenesis: too much active Yorkie accelerates myofiber compaction, while too little active Yorkie blocks it.

To further investigate how the STRIPAK complex may regulate the Hippo pathway in flight muscles, we investigated Hippo protein localisation in relation to Slmap protein. We find that the available Hippo antibody does give specific staining in flight muscles at 32 hr APF, which is strongly reduced in the *hippo* knock-down flight muscle (*Figure 4—figure supplement 1D*). Detailed imaging revealed that Hippo localises in a dotted pattern in flight muscles at 32 hr APF, with the dots often being in proximity to Slmap-positive structures; however, we found little overlap of both (*Figure 4—figure supplement 1E*).

Hence, we wanted to corroborate that the STRIPAK complex directly regulates the Hippo pathway in flight muscles with additional genetic evidence and used a recently characterised *hippo* construct (*hippo[4A/431T]*), which lacks the four auto-phosphorylation sites in Hippo required to bind to the STRIPAK phosphatase complex via Slmap. This Hippo[4A/431T] protein cannot be dephosphorylated on regulatory T195 by STRIPAK and thus is constitutively active (*Zheng et al., 2017*). Interestingly, expression of *hippo[4A/431T]* in muscle after *Mef2*-GAL4 driven flip-out also resulted in a muscle fiber compaction defect at 32 hr APF and muscle atrophy thereafter, phenocopying the STRIPAK and *yorkie* loss-of-function phenotypes (*Figure 4C*). Taken all these data together, we conclude that Dlg5 and members of the STRIPAK complex are key regulators of the Hippo pathway, which controls the developmental timing of flight muscle morphogenesis in *Drosophila*.

## The Hippo pathway is required post-mitotically in flight muscle fibers

Indirect flight muscles are formed by fusion of several hundred myoblasts until 24 hr APF (*Weitkunat et al., 2014*). These myoblasts emerge during embryonic development and proliferate extensively during larval stages (*Bate et al., 1991*; *Gunage et al., 2014*; *Roy and VijayRaghavan, 1998*). As knock-down of *Dlg5*, STRIPAK complex, and Hippo pathway members with *Mef2*-GAL4 results in flightless animals and not in lethality (except for *warts*, see *Figure 4—figure supplement 1A*), it is unlikely that general myoblast proliferation during larval stages is affected, which would result in defects of all adult muscles. However, since *Mef2*-GAL4 is already active during larval stages, we wanted to exclude that the observed muscle phenotypes are caused by myoblast proliferation defects during larval stages. Hence, we conditionally activated GAL4 only during pupal stages using temperature sensitive GAL80 (GAL80ts, see Materials and methods) (*McGuire et al., 2003*) and quantified myoblast fusion rates by counting the nuclei of DLM4. We found comparable numbers of nuclei at 24 hr APF ruling out a major contribution of myoblast proliferation or fusion defect to the phenotype (*Figure 5—figure supplement 1A,B*).

Importantly, these GAL80ts *Mef2*-GAL4 *Dlg5* and *yorkie* knock-down muscles do display the same fiber compaction defect as observed with *Mef2*-GAL4 resulting in longer but thinner fibers with grossly comparable volumes at 24 hr APF (*yorkie-IR* is slightly smaller) (*Figure 5A,B*). These fibers do not compact at 32 hr APF and undergo atrophy leading to no remaining fibers at 48 hr or 90 hr APF (*Figure 5C,D*), phenocopying the constitutive knock-down of *Dlg5* or *yorkie*. Conversely, conditional expression of *yorkie-CA* during pupal stages results in premature compaction at 24 hr APF and very short fibers at 32 hr APF that grow to disorganised fibers at 90 hr APF (*Figure 5A–D*). These phenotypes resemble the constitutive *Mef2*-GAL4-driven phenotypes demonstrating a role for *Dlg5* and *yorkie* in muscle fibers during pupal stages.

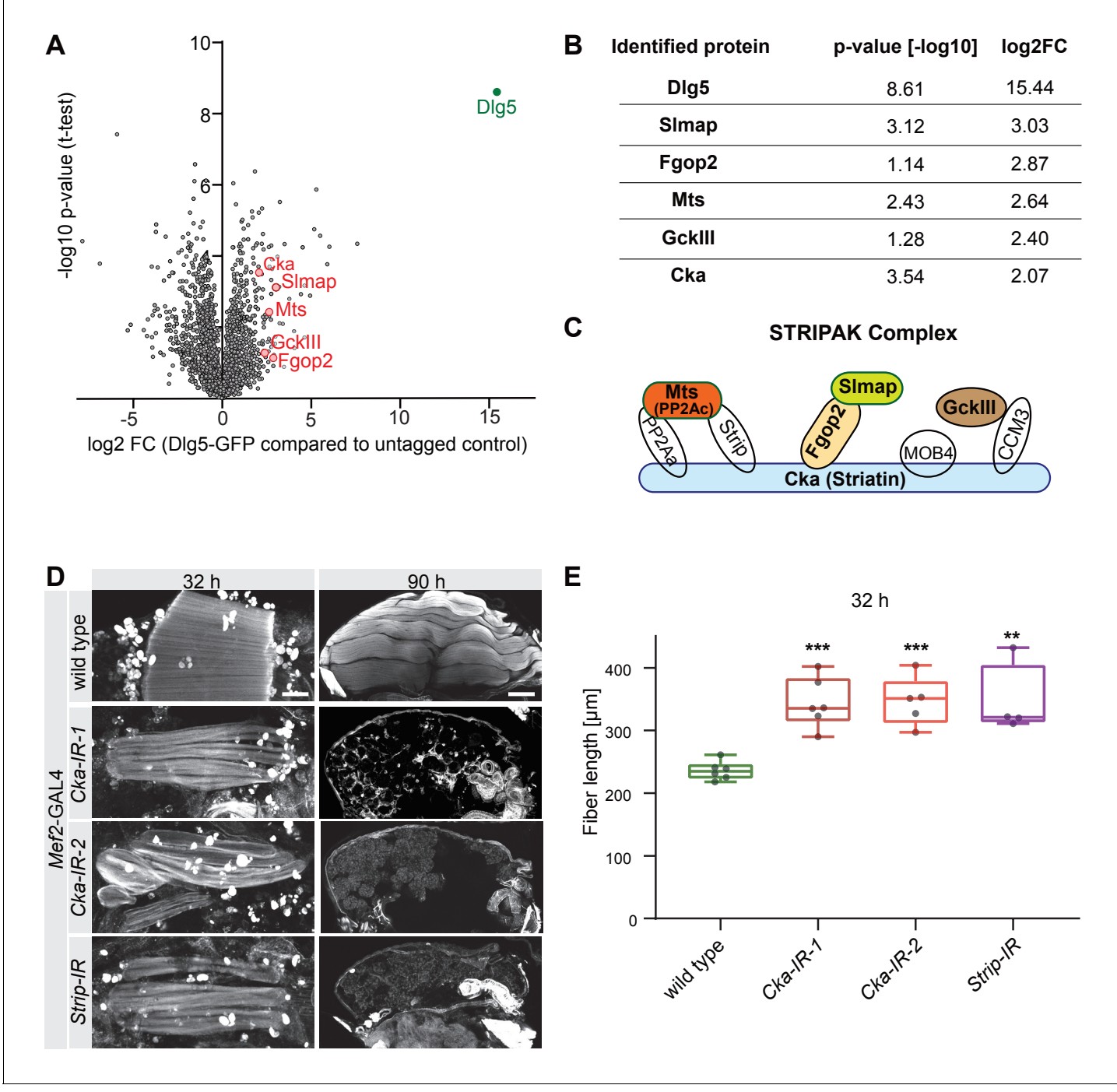

**Figure 3.** Dlg5 binds to the STRIPAK complex. (**A**) Volcano plot showing proteins enriched in GFP pull-down from *Mef2*-GAL4, *UAS-Dlg5-GFP* 24 hr - 48 hr after puparium formation (APF) pupae. Y-axis shows statistical significance (-log10) and x-axis the $\log_2$fold change (FC) compared to *Mef2*-GAL4 control (see ***Supplementary file 1***). STRIPAK complex components are highlighted in red. (**B**) Table showing the $\log_2$fold change and p-values of the selected STRIPAK complex components. (**C**) Simplified schematic of the STRIPAK complex (adapted from ***Duhart and Raftery, 2020***). Proteins identified in the Dlg5-GFP pull-down are coloured. (**D**) Flight muscles at 32 hr and 90 hr APF of wild type, *Cka* (independent RNAi lines *IR-1* and *2*) or *Strip* knockdown genotypes stained for actin. Note the compaction defect at 32 hr APF followed by muscle atrophy. Scale bars represent 50 μm for 32 hr and 100 μm for 90 hr APF. (**E**) Box plot showing dorsal-longitudinal flight muscle (DLM) fiber length at 32 hr APF of the indicated genotypes. Each dot is the average from one pupa. Student's t-test, *** p-value<0.001, ** p-value<0.01.

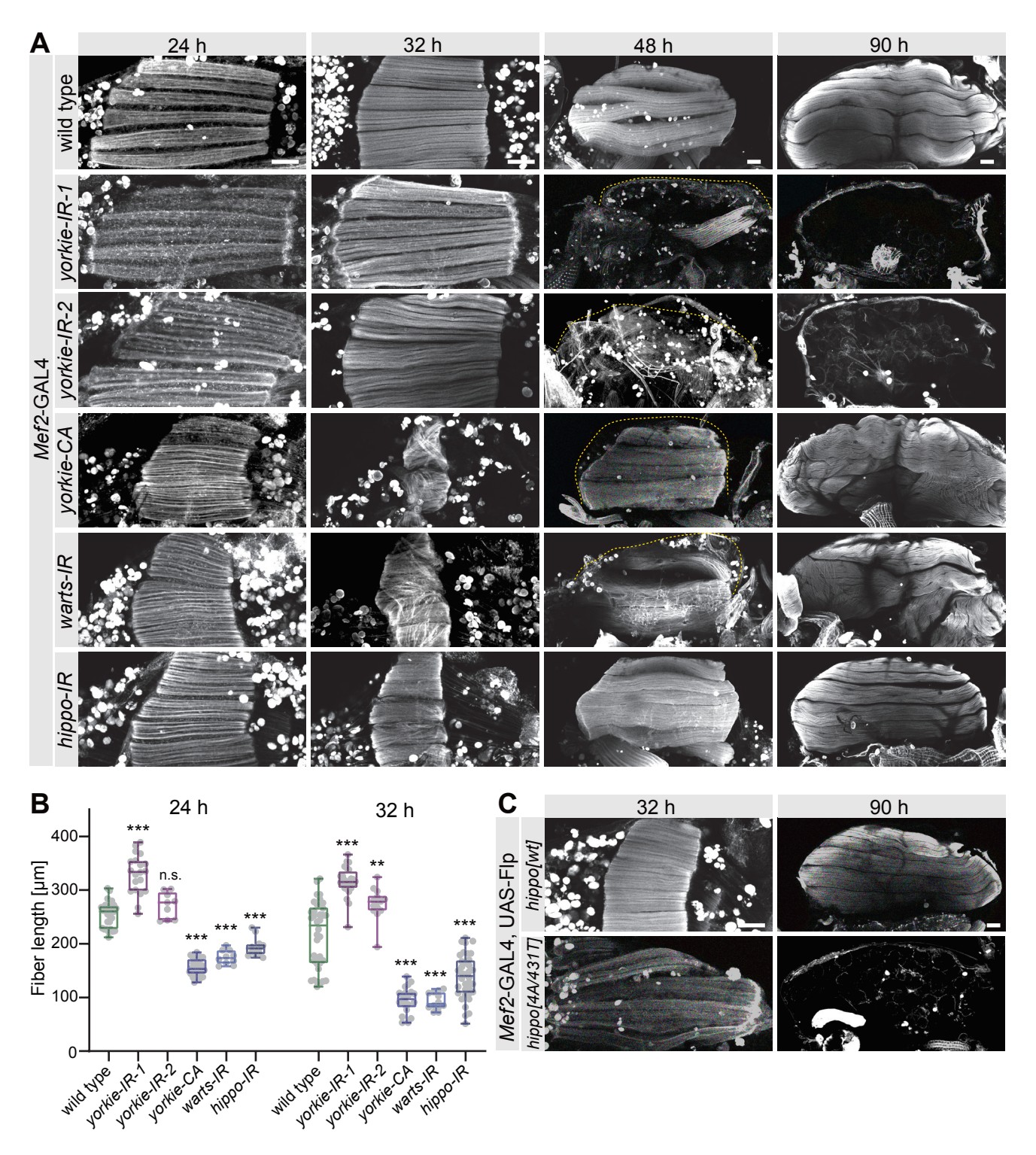

**Figure 4.** The Hippo pathway regulates muscle morphogenesis. (**A**) Flight muscles at 24 hr, 32 hr, 48 hr, and 90 hr after puparium formation (APF) from wild type, *yorkie* knock-down (independent RNAi lines *IR-1* and *2*), *yorkie-CA*, as well as *warts* and *hippo* knockdown genotypes stained for actin. The dotted lines highlight the cuticle. Note the too long *yorkie-IR* muscles but too short *yorkie-CA*, *warts-IR*, and *hippo-IR* muscles at 24 hr and 32 hr APF. (**B**) Box plot showing muscle fiber length at 24 hr and 32 hr APF of the indicated genotypes. Student's t-test, *** p-value<0.001, **p value<0.01. (**C**) Flight muscles at 32 hr and 90 hr APF from pupae expressing either wild-type *hippo* or Slmap-binding-deficient *hippo[4A/431T]* (under control of the milder tubulin promoter). The FRT stop cassette is removed by muscle-specific *Mef2*-GAL4-driven Flp recombinase. All scale bars represent 50 µm.
*Figure 4 continued on next page*

*Figure 4 continued*

The online version of this article includes the following figure supplement(s) for figure 4:

**Figure supplement 1.** The Hippo pathway is required for muscle function.

To further corroborate a post-mitotic role of the Hippo pathway in muscle fibers, we over-expressed *hippo* with the strong, strictly post-mitotic flight muscle specific driver *Act88F*-GAL4, which is only active after myoblast fusion (*Bryantsev et al., 2012*; *Spletter et al., 2018*). This post-mitotic over-expression of *hippo* resulted in flight muscle compaction defects at 32 hr APF and muscle atrophy at 48 hr APF (*Figure 5E*). Together, these data demonstrate that the Hippo pathway and its regulator Dlg5 are required post-mitotically in flight muscle fibers for the correct timing of morphogenesis.

## The Hippo pathway regulates post-mitotic muscle fiber growth

The wild-type flight muscle fibers grow in volume from 24 hr to 48 APF (see *Figure 1*), while *yorkie* or *Dlg5* knock-down fibers undergo atrophy after 32 hr APF. As the Yorkie activity is known to suppress apoptosis in epithelial tissues (*Harvey and Tapon, 2007*) we asked if we could rescue fiber atrophy by over-expressing the apoptosis inhibitor Diap1. Indeed, over-expression of Diap1 using *UAS-Diap1* but not of a control construct (*UAS-GFP-Gma*) during pupal stages in GAL80ts *Mef2*-GAL4 (hereafter abbreviated as GAL80ts) *yorkie* or *Dlg5* knock-down fibers substantially rescues fiber atrophy, often resulting in the normal number of six muscle fibers at 48 hr APF (*Figure 6A*, *Figure 6—figure supplement 1A*). Furthermore, over-expression of the exogenous apoptosis inhibitor p35 (*Clem et al., 1991*) also results in six muscle fibers at 48 hr APF (*Figure 6—figure supplement 1B,C*). This demonstrates that apoptosis contributes to flight muscle fiber atrophy in *yorkie* and *Dlg5* knock-down muscles.

Presence of muscle fibers at 48 hr APF enabled us to quantitatively investigate the role of the Hippo pathway during the post-mitotic muscle fiber growth. As in *Figure 5*, we used digital cross-sections of large confocal stacks to quantify the cross-sectional area of DLM4 and together with the fiber length calculated the fiber volume (*Figure 6A–C*). Similar to what we showed in *Figure 5*, GAL80ts *Diap1*-expressing control fibers have a comparable volume to GAL80ts *Diap1*-expressing *yorkie* or *Dlg5* knock-down fibers at 24 hr APF (*Figure 6A–C*), showing that they start into the muscle growth phase with comparable sizes.

However, until 32 hr APF, *Diap1* control muscles increase their cross-sectional area followed by growth in length until 48 hr APF to increase their volume about fourfold within 24 hr (*Figure 6A–C*). Interestingly, GAL80ts *Diap1*-expressing *yorkie* or *Dlg5* knock-down muscles fail to normally increase their cross-sectional area at 32 hr and 48 hr APF, thus resulting in smaller volumes at 48 hr APF (*Figure 6A–C*). Not all these thin muscles do survive until 90 hr APF despite over-expression of the apoptosis inhibitors Diap1 or p35 (*Figure 6A* and *Figure 6—figure supplement 1B*). Taken together, these data provide strong evidence that the Hippo pathway and its transcriptional co-activator Yorkie are required to enable normal post-mitotic growth of flight muscle fibers, likely by regulating the developmental timing of muscle morphogenesis.

## The Hippo pathway is essential for myofibrillogenesis triggering muscle growth

To investigate the molecular mechanism of the muscle growth defect in detail, we quantified myofibrillogenesis. Control *Diap1*-expressing muscles have assembled immature myofibrils at 32 hr APF (*Weitkunat et al., 2014*). These immature myofibrils are continuous and thus can be easily traced throughout the entire field of view (*Figure 7A,B*, *Figure 7—figure supplement 1A*). In contrast, GAL80ts *Diap1*-expressing *yorkie* or *Dlg5* knock-down muscles fail to properly assemble their myofibrils at 32 hr APF resulting in only short myofibril traces (*Figure 7A,B*, *Figure 7—figure supplement 1A*). Concomitant with the myofibril assembly defect, we also found that the spacing of the nuclei is defective. In control muscle fibers the nuclei are present mainly as single rows located between myofibril bundles, whereas in *yorkie* and *Dlg5* knock-down muscles they form large centrally located clusters (*Figure 7—figure supplement 1B*). This indicates that at 32 hr APF, Hippo signalling is required within the muscle fibers to trigger proper myofibril assembly and nuclear positioning.

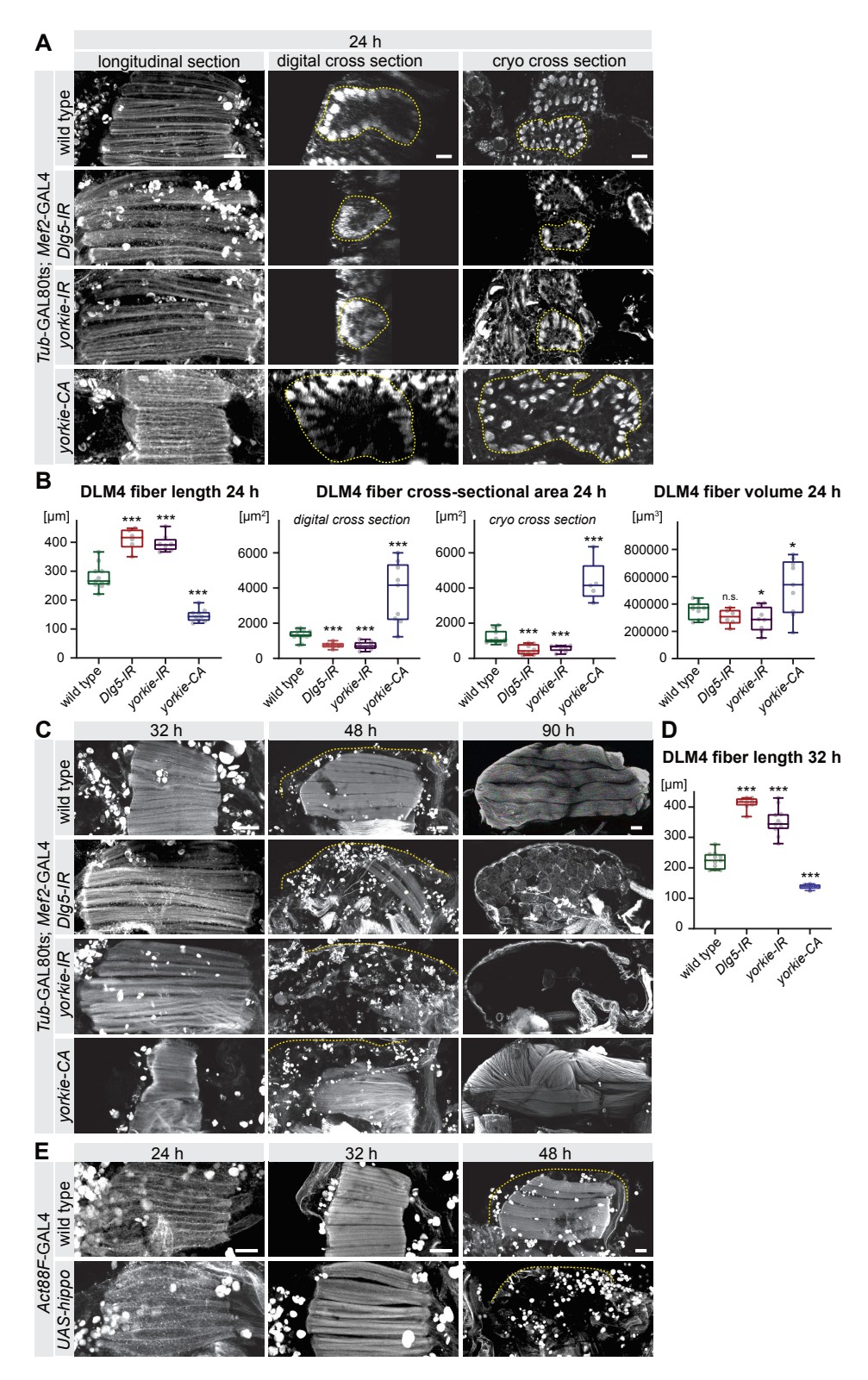

**Figure 5.** The Hippo pathway is important for post-mitotic muscle development. (**A–D**) Developing flight muscles at 24 hr, 32 hr, 48 hr, and 90 hr after puparium formation (APF) from wild type, *Dlg5-IR*, *yorkie-IR*, and *yorkie-CA Tub*GAL80ts *Mef2*-GAL4 pupae, in which GAL4 activity was restricted to pupal stages (shift to 31°C at 0 hr APF). (**A**) Longitudinal sections of all flight muscles, as well as digital and cryo cross-sections of dorsal-longitudinal flight muscle 4 (DLM4) at 24 hr APF. Dotted lines highlight the fiber area. (**B**) Box plot displaying DLM4 muscle fiber length, fiber cross-sectional area

*Figure 5 continued on next page*

*Figure 5 continued*

from digital and cryo cross-sections as well as the calculated DLM4 fiber volume at 24 hr APF (calculated by multiplying length with digital cross-sectional area for each pupa). Student's t-test, *** p-value<0.001, * p-value<0.05. (**C**) Flight muscles at 32 hr, 48 hr, and 90 hr APF. Note the too long muscles in *Dlg5-IR, yorkie-IR* at 32 hr APF followed by muscle atrophy and the too short *yorkie-CA* fibers at 32 hr that develop disorganised fibers at 90 hr APF. (**D**) Box plot illustrating the average muscle fiber length at 32 hr APF. Student's t-test, *** p-value<0.001. (**E**) Control and *Act88F*-GAL4 *UAS-hippo* flight muscles at 24 hr, 32 hr and 48 hr APF. Note the induced fiber compaction defect followed by the muscle atrophy. All scale bars represent 50 µm for longitudinal sections and 10 µm for cross-sections.

The online version of this article includes the following figure supplement(s) for figure 5:

**Figure supplement 1.** The Hippo pathway is not required for myoblast fusion.

The myofibril defect becomes even more pronounced at 48 hr APF when control myofibrils have matured and sarcomeres are easily discernible (*Figure 7A*), while no organised sarcomeres are present in GAL80ts *yorkie* and *Dlg5* knock-down muscles at 48 hr APF and myofibril traces remain short (*Figure 7A,B*, *Figure 7—figure supplement 1A*). Furthermore, cryo cross-sections revealed that not only the cross-sectional area but also the total number of myofibrils is strongly reduced in GAL80ts *yorkie* and *Dlg5* knock-down muscles compared to control (*Figure 7C,D*, *Figure 7—figure supplement 1C*). As myofibril number is determined at the assembly stage at 32 hr APF in wild type (*Spletter et al., 2018*) these data demonstrate that the Hippo pathway controls both the morphological quality of the myofibrils at the assembly and maturation stages as well as their quantity. As myofibrils occupy most of the muscle fiber space, their reduced amount likely causes the reduced muscle size in *Dlg5* or *yorkie* knock-down fibers.

To further substantiate that the amount and quality of myofibrils are important for normal muscle growth, we aimed for an independent way of changing myofibril number and assessing muscle fiber size. It had been shown that mechanical tension in flight muscles is a prerequisite for normal myofibrillogenesis (*Weitkunat et al., 2014*). Upon hypomorphic knock-down of the muscle attachment protein Kon-tiki (*kon*), muscle-tendon attachment is compromised, hence mechanical tension build-up is perturbed and only fewer myofibrils can assemble (*Weitkunat et al., 2014*). Consistent with our results after *yorkie* or *Dlg5* knock-down we find that *kon* knock-down also results in a fiber compaction defect, caused by reduced mechanical tension at 32 hr APF (*Figure 7—figure supplement 1D*), and as a consequence in a strongly reduced fiber cross-sectional area and volume at 48 hr APF (*Figure 7—figure supplement 1D–F*). Taken together, these data strongly suggest that both mechanical tension and regulated Yorkie activity are required for normal myofibril morphogenesis to support muscle fiber growth.

## Yorkie is a transcriptional co-regulator in muscle fibers

It was recently shown that the transcription of most sarcomere key components is tightly regulated starting shortly before myofibril assembly and being strongly boosted during myofibril maturation (*Spletter et al., 2018*). Thus, we reasoned that Yorkie activity may be involved in this transcriptional regulation step to control the timing of myofibrillogenesis. However, it had also been recently shown that Yorkie can regulate myosin contractility directly at the cell membrane without entering into the nucleus (*Xu et al., 2018*). As we have thus far failed to unambiguously locate Yorkie protein in muscle fibers during development, we used genetic tools to address this important point. To test whether Yorkie may play a role outside of the nucleus, we manipulated Yorkie levels by over-expressing different Yorkie variants post-mitotically using *Act88F*-GAL4 and investigated the consequences at 24 hr and 32 hr APF. Over-expression of either Yorkie-CA, whose import into the nucleus is uncoupled from the Hippo pathway, or wild-type Yorkie, whose nuclear import is regulated by Hippo, both result in premature muscle fiber compaction at 24 hr APF, with seemingly normal actin filaments, phenocopying the Gal80ts condition (*Figure 7E*). Strikingly, the muscle fiber hyper-compaction at 32 hr APF coincides with a chaotic organisation of the myofibrils, with many myofibrils not running in parallel but in various directions (*Figure 7E*, *Figure 7—figure supplement 2A*). This suggests that the hyper-compaction phenotype upon Yorkie over-expression is likely caused by uncontrolled and premature force production of the chaotically assembling myofibrils.

In contrast, over-expression of a membrane-anchored myristoylated form of Yorkie, which has been shown to activate myosin contractility at the epithelial cell cortex without going into the

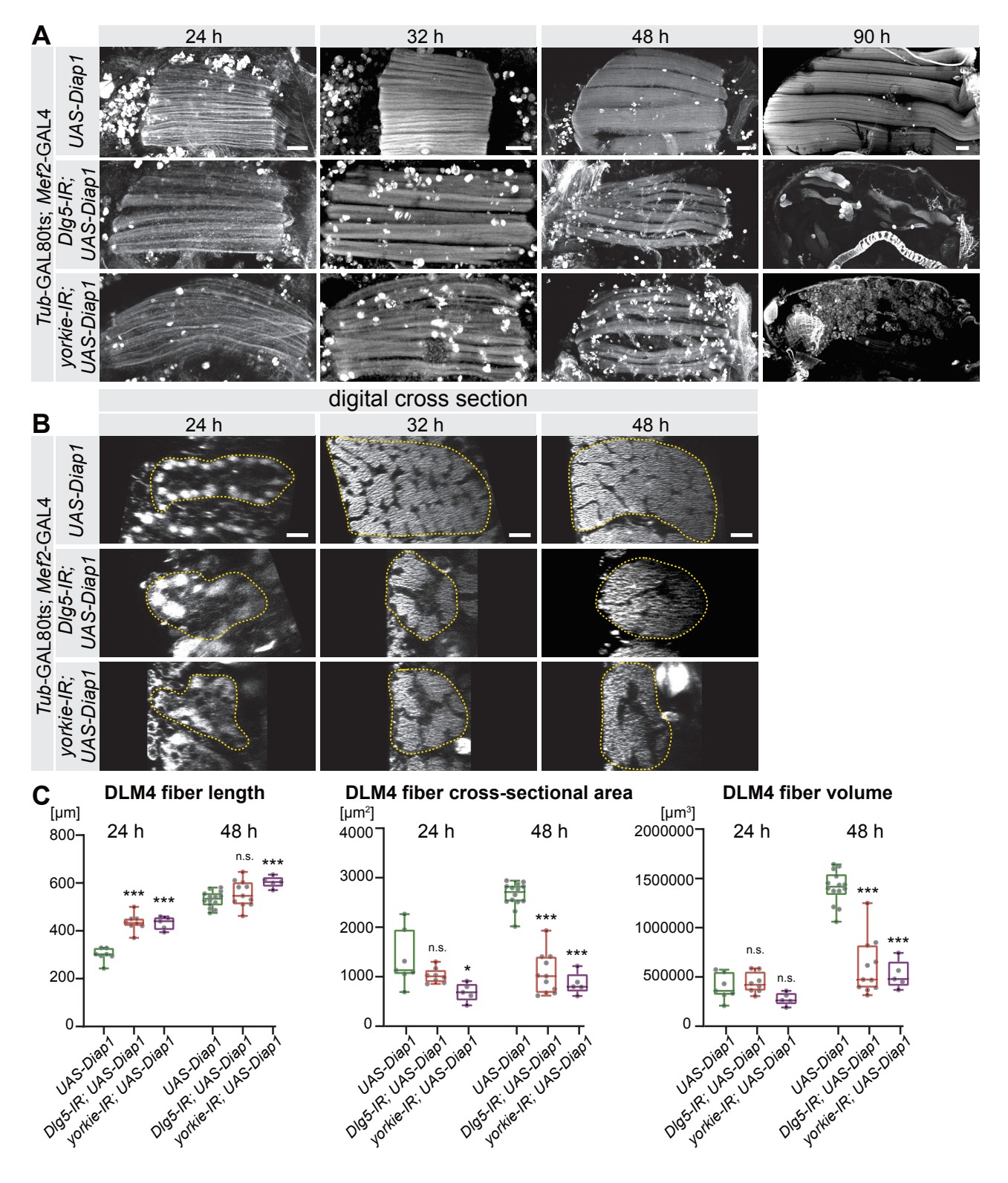

**Figure 6.** The Hippo pathway is essential for post-mitotic muscle fiber growth. (**A, B**) Longitudinal sections of all dorsal-longitudinal flight muscles (DLM) at 24 hr, 32 hr, 48 hr, and 90 hr after puparium formation (APF) (**A**) and digital cross-sections of DLM4 at 24 hr, 32 hr, and 48 hr APF (**B**) from wild type, *Dlg5-IR and yorkie-IR Tub*GAL80ts *Mef2*-GAL4 *UAS-Diap1* (shifted to 31°C at 0 hr APF) stained for actin. Scale bars represent 50 µm for longitudinal sections and 10 µm for cross-sections. In B dotted lines highlight the fiber areas. (**C**) Box plots showing DLM4 fiber length, digital cross-

*Figure 6 continued on next page*

*Figure 6 continued*
sectional area, and volume (calculated by multiplying length with digital cross-sectional area for each pupa) at 24 hr and 48 hr APF. Student's t test, *** p-value<0.001, * p-value<0.05.
The online version of this article includes the following figure supplement(s) for figure 6:

**Figure supplement 1.** Apoptosis block and muscle growth.

nucleus (*Xu et al., 2018*), does not result in premature muscle fiber compaction at 24 hr APF. Furthermore, these muscles display normally oriented parallel myofibrils at 32 hr APF (*Figure 7E*, *Figure 7—figure supplement 2A*). These results indicate that the observed myofibril and fiber compaction defects are caused by a transcriptional response of Yorkie in the nucleus.

This interpretation is corroborated by loss-of-function data of the transcriptional activator *scalloped* (*sd*), which is the essential transcriptional co-factor of Yorkie in the nucleus. Knock-down of *scalloped* results in severe muscle atrophy and no remaining muscles at 90 hr APF (*Figure 7—figure supplement 2B*). Together, these genetic data suggest that Hippo signalling regulates Yorkie phosphorylation and thus its nuclear entry to trigger a transcriptional response that controls myofibril development and muscle fiber growth.

## The Hippo pathway controls expression of key sarcomere components

To investigate the transcriptional role of the Hippo pathway during flight muscle development, we performed muscle-specific transcriptomics of wild-type flight muscles compared to different *yorkie* 'loss-of-function' (*Dlg5-IR*, *Slmap-IR,* and *yorkie-IR*) and *yorkie* 'gain-of-function' (*yorkie-CA and hippo-IR*) conditions. We dissected flight muscles from 24 hr and 32 hr APF, isolated RNA and applied a sensitive 3-prime end mRNA sequencing method (BRB-seq) (*Alpern et al., 2019*), which handles small amounts of mRNA (see Materials and methods). We found a clustering of biological replicates and similar genotypes using principal component analysis and comparable read count distributions across all samples (*Figure 8—figure supplement 1*). This verifies BRB-seq as a reliable method to quantitatively compare gene expression from small amounts of developing muscle tissue across multiple samples.

We applied the selection criteria log2FC > 1 and adjusted p-value<0.05 to identify differentially expressed genes compared to wild type (*Supplementary file 2*). Applying FlyEnrichr (*Kuleshov et al., 2016*) on the differential data sets, we found a strong enrichment for muscle and, in particular, for sarcomere and myofibril Gene Ontology terms (GO-terms) in the differentially expressed genes of all three *yorkie* 'loss-of-function' muscle genotypes (*Dlg5-IR*, *Slmap-IR,* and *yorkie-IR*) at 24 hr APF (*Figure 8A*, *Supplementary file 3*). Importantly, expression of many core sarcomeric components, including both titin homologs *sallimus* (*sls*) and *bent* (*bt*), *Myosin heavy chain* (*Mhc*), *Myofilin* (*Mf*), *Paramyosin* (*Prm*), tropomyosins (*Tm1*, *Tm2*), flight-muscle-specific actin (*Act88F*) and *Obscurin* (*Unc-89*), as well as sarcomere dynamics regulators, including myosin phosphatase (*Mbs*), a flight muscle formin (*Fhos*) and the spektraplakin *shortstop* (*shot*) are consistently reduced in *yorkie* 'loss-of-function' muscle genotypes at 24 hr APF (*Figure 8B*). Furthermore, expression of mRNAs coding for proteins linking the nuclei to the cytoskeleton, such as the Nesprin family members *klar* and *Msp300*, are also strongly reduced (*Figure 8B*), which may explain the observed nuclei positioning defect in *yorkie* and *Dlg5* knock-down muscles (*Figure 7—figure supplement 1B*). *Msp300* and *Prm* are amongst the only six genes that are significantly downregulated in all three loss-of-function conditions at 24 hr APF (*Supplementary file 2*). This strongly suggests that nuclear entry of Yorkie contributes to the transcriptional induction of sarcomeric protein coding genes as well as genes important to link the nuclei to the sarcomeres. This transcriptional induction was shown to precede sarcomere assembly (*Spletter et al., 2018*) and thus may provide a molecular explanation of the observed flight muscle compaction and myofibril assembly defects of *Dlg5-IR*, *Slmap-IR,* and *yorkie-IR* muscles.

To complement the *yorkie* loss-of-function conditions, we also performed BRB-seq transcriptomics comparing *yorkie* 'gain-of-function' conditions (*yorkie-CA* and *hippo-IR*) to control. While we found few significantly differentially expressed genes at 24 hr APF (including an induction of the transcriptional activator *scalloped*) (*Supplementary file 2*), we identified many significant changes in mRNA expression in *yorkie-CA* and *hippo-IR* myofibers at 32 hr APF (*Supplementary file 2*).

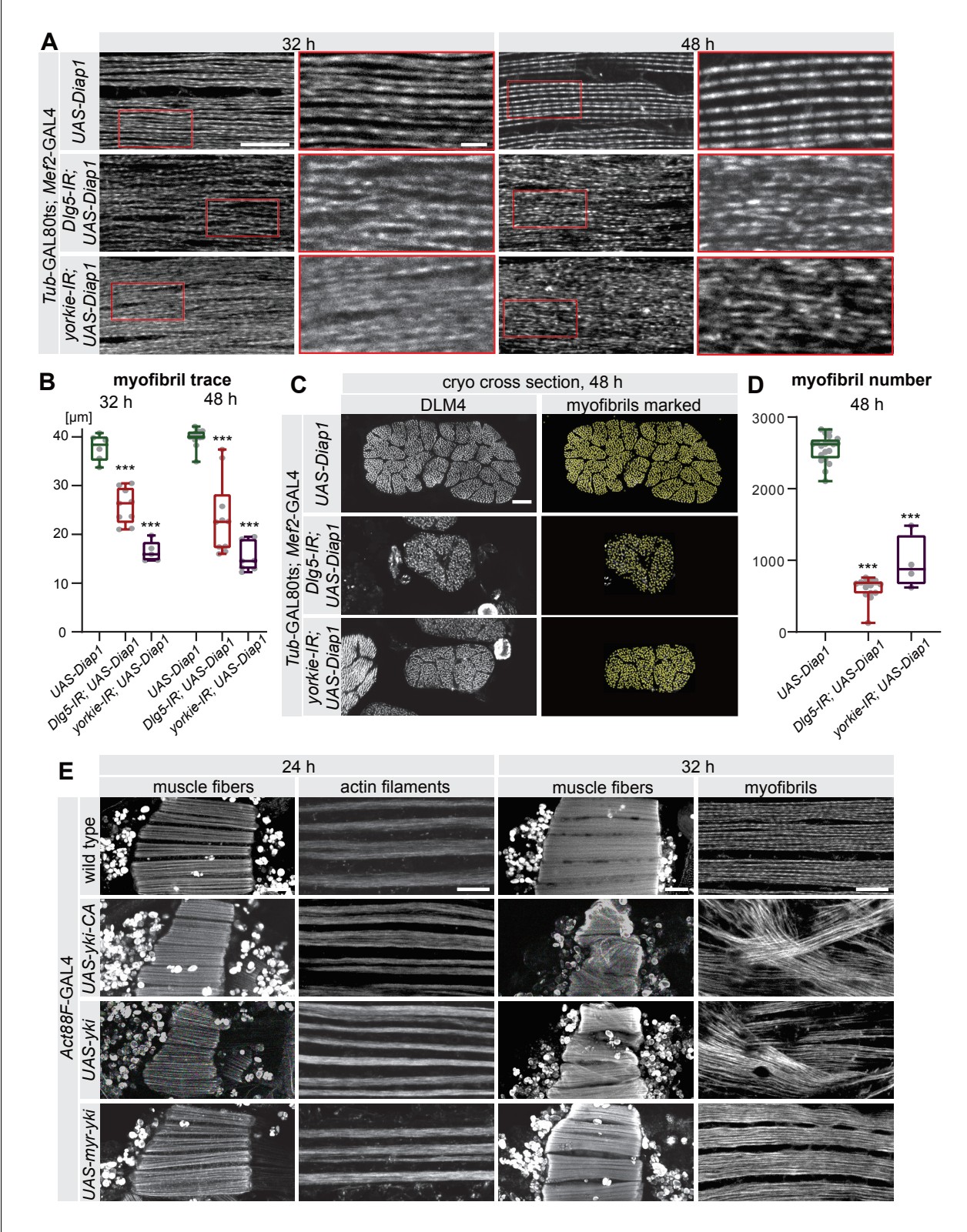

**Figure 7.** The Hippo pathway is essential for myofibrillogenesis. (**A**) Myofibrils visualised by phalloidin from control, *Dlg5-IR,* and *yorkie-IR UAS-Diap1 Tub*GAL80ts *Mef2*-GAL4 muscles at 32 hr and 48 hr after puparium formation (APF; shifted to 31°C at 0 hr APF). The red-boxed areas are magnified. Note the less regular myofibril pattern of *Dlg5-IR* and *yorkie-IR* muscles at 32 hr APF that makes it hard to trace an individual myofibril (see **Figure 7— figure supplement 1A**). Even at 48 hr APF, myofibrils from *Dlg5-IR* and *yorkie-IR* are hard to trace continuously. Scale bars represent 10 μm in the *Figure 7 continued on next page*

*Figure 7 continued*

overviews and 2 µm in the zoomed red boxes. (**B**) Box plot of traced myofibril length in a 40 × 20 × 2.5 µm volume (see *Figure 7—figure supplement 1A*). Student's t test, *** p-value<0.001. (**C**) Cryo cross-sections of dorsal-longitudinal flight muscle 4 (DLM4) from control, *Dlg5-IR* and *yorkie-IR UAS-Diap1 Tub*GAL80ts *Mef2*-GAL4 muscles at 48 hr APF (shifted to 31°C at 0 hr APF). Yellow dots represent the myofibrils recognised by the MyofibrilJ plug-in to automatically count the number of myofibrils per DLM4 fiber (*Spletter et al., 2018*). Scale bar represents 10 µm. (**D**) Box plot of myofibril number in DLM4 of indicated genotypes at 48 hr APF. Student's t test, *** p-value<0.001. (**E**) Flight muscle and myofibril morphologies of muscles expressing *yorkie-CA*, *yorkie,* or *myr-yorkie* under the control of post-mitotic *Act88F*-GAL4 at 24 hr and 32 hr APF. Scale bars represent 50 µm in muscle fiber images and 10 µm in myofibril images.

The online version of this article includes the following figure supplement(s) for figure 7:

**Figure supplement 1.** Myofibril tracing, nuclei positions in *Dlg5-IR* and *yorkie-IR* muscles and muscle growth in *kon-IR* fibers.
**Figure supplement 2.** Yorkie is active in the nucleus.

Strikingly, sarcomeric core components and their regulators are amongst the top upregulated genes on both lists (*Figure 8B*, *Supplementary file 2*). Consequently, GO-term analysis of the differentially expressed genes identified a strong enrichment for sarcomere and myofibril GO-terms (*Figure 8A*, *Supplementary file 3*). In addition to sarcomeric genes, genes important for myofibril attachment at muscle-tendon junctions, including the integrin attachment complex members *kon*, *if*, *CAP*, *by*, and *Kank* are upregulated in both gain-of-function genotypes (*Figure 8B*). Furthermore, we found an upregulation of Hippo signalling regulators *mask*, which is required for efficient nuclear import of Yki (*Sidor et al., 2019*), and of the negative regulator *wts* (*Figure 8B*). This demonstrates that regulated Hippo activity is required to control Yorkie in order to tune expression of mRNAs coding for sarcomeric and myofibril attachment proteins.

During the stage of myofibril assembly, the mitochondria morphology changes and the expression of mitochondrial genes increase (*Avellaneda et al., 2020*; *Spletter et al., 2018*). Consistently, we found an induction of mitochondria dynamics and protein import regulators (Opa1, Tom40) in *hippo-IR and yorkie-CA* myofibers at 32 hr APF as well as a consistent upregulation of mRNAs coding for respiratory chain components, including the F1F0 ATP synthase complex (complex V) subunit *blw*, the NADH dehydrogenase (ubiquinone) subunit *ND-75* and the Ubiquinol-cytochrome c reductase subunit *UQCR-C2*, which are all required to boost ATP production during muscle fiber growth (*Figure 8A,B*, *Supplementary file 2*). Taken together, these data strongly suggest that the Hippo pathway regulates the correct expression dynamics of many key muscle components, most prominently mRNAs coding for core sarcomeric and mitochondrial proteins to enable myofibril assembly and mitochondrial maturation.

## Yorkie controls sarcomeric protein dynamics

The most prominent phenotypes of the *yorkie* 'loss-of-function' group (*Dlg5-IR*, *Slmap-IR* and *yorkie-IR*) are defective muscle fiber compaction and severe myofibril assembly defects at 32 hr APF. As the core sarcomeric proteins actin and myosin are required to assemble myofibrils and build-up mechanical tension (*Loison et al., 2018*; *Weitkunat et al., 2014*), we chose to quantify protein levels of the major actin isoform in flight muscles Actin88F (Act88F) as well as the only *Drosophila* muscle Myosin heavy chain (Mhc). For both, we used GFP fusion proteins expressed under endogenous control (*Sarov et al., 2016*) and thus avoiding the variations often seen in antibody stainings (see Materials and methods). Consistent with the transcriptomics data, we found a mild reduction of Mhc and Act88F protein levels in *yorkie* knock-down muscles at 24 hr APF, which became more pronounced at 32 hr APF (*Figure 9A–C*). Furthermore, we have investigated protein levels for the myosin-binding protein Paramyosin (Prm) and the titin homolog Sls also using GFP fusion proteins. We found a reduction of Sls-GFP at 24 hr APF and a severe reduction of Prm-GFP at 32 hr APF in *yorkie* knock-down muscles (*Figure 9—figure supplement 1A–C*). Together, these data are consistent with the myofibril assembly defects found in *yorkie* knock-down muscles at 32 hr APF.

Surprisingly, constitutive activation of *yorkie* (*yorkie-CA*) results in a boost of Mhc-GFP, Act88F-GFP, Sls-GFP, and Prm-GFP protein expression already at 24 hr APF, which is maintained at 32 hr APF (*Figure 9A–C*, *Figure 9—figure supplement 1A–C*). This increased acto-myosin and titin expression may provide the molecular explanation of the premature compaction phenotype seen in *yorkie-CA* muscles at 24 hr APF. Despite being expressed at high levels already at 24 hr APF, the proteins fail to prematurely assemble into well-organised periodic myofibrils in *yorkie-CA*

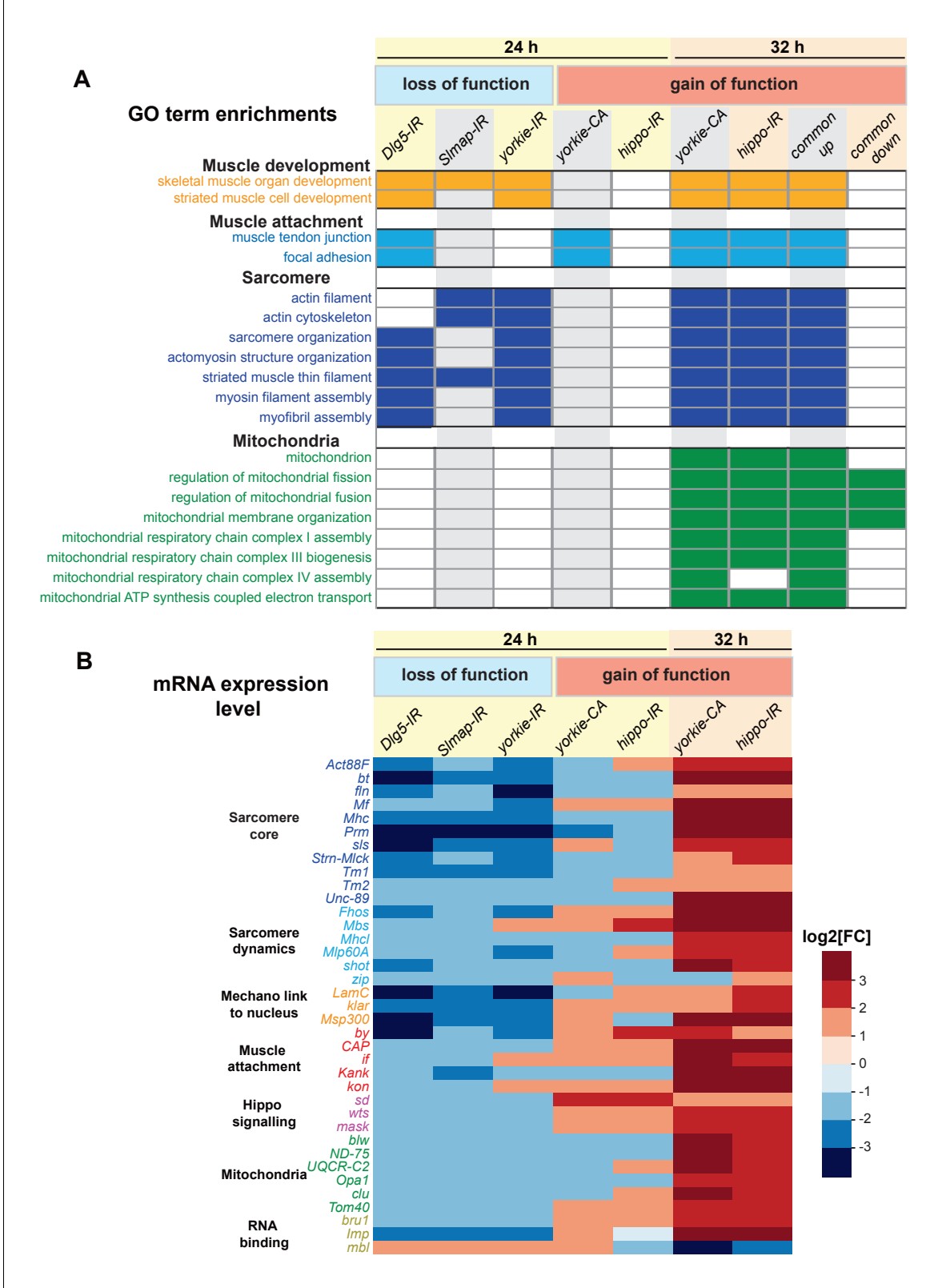

**Figure 8.** Yorkie transcriptionally controls sarcomeric and mitochondrial genes expression. (**A**) Gene ontology (GO)-term enrichments in genes lists that are significantly changed in the various loss (*Dlg5-IR, Slmap-IR, yorkie-IR*) and gain-of-function (*yorkie-CA, hippo-IR*) yorkie conditions at 24 hr and 32 hr after puparium formation (APF) compared to wild-type controls (see **Supplementary file 3**). Note the strong enrichment of sarcomere related GO-terms in the 24 hr APF loss-of-function and the 32 hr APF gain-of-function condition. Thirty-two hr APF gain of function is also strongly enriched for
*Figure 8 continued on next page*

*Figure 8 continued*

mitochondrial GO-terms. (**B**) Plot displaying the log2-fold change of transcript levels for individual genes of the above genotypes compared to control (up in red, down in blue). Note the strong downregulation of the sarcomeric genes in the 24 hr APF loss-of-function conditions, in particular the titin homologs *bt* and *sls*, as well as *Mhc, Prm* and *Act88F*.

The online version of this article includes the following figure supplement(s) for figure 8:

**Figure supplement 1.** PCA analysis and BRB sequencing read counts.

---

(*Figure 9A,B*, *Figure 9—figure supplement 1A,B*). Furthermore, the myofibrils present in *yorkie-CA* at 32 hr APF are less regularly organised compared to control, with actin showing less pronounced periodicity (*Figure 7—figure supplement 2A*). These ectopic localisation patterns of actin, myosin, and titin may explain muscle hyper-compaction at 32 hr APF and the chaotic arrangement of myofibrils in *yorkie-CA* 90 hr APF (see *Figure 5C*). Taken together, these data provide strong evidence that the Hippo pathway and its transcriptional regulator Yorkie contribute to the timing of myofibril assembly by regulating correct timing and levels of sarcomeric protein expression.

## Discussion

### The Hippo pathway and post-mitotic muscle fiber growth

Muscle fibers are often enormously large cells, which are densely packed with force producing contractile filaments and ATP-producing mitochondria (*Willingham et al., 2020*). Thus, it appears logical that myofibril and mitochondria content primarily contribute to the size of an individual muscle fiber. To explore the link between myofibrillogenesis and muscle fiber growth, we used the largest *Drosophila* muscle cells, the indirect flight muscles, which are formed by fusion of several hundred myoblasts per muscle fiber and grow more than 10 times in volume after fusion in less than 72 hr. We provide strong genetic evidence that the regulation of Yorkie nuclear activity by the Hippo pathway is essential to allow post-mitotic flight muscle growth. Loss of function of Yorkie or one of the upstream STRIPAK complex members Slmap, Strip, and Cka, as well as Dlg5, that regulate Hippo activity, all show the same phenotype: muscle atrophy during the post-mitotic muscle growth phase. Atrophy can be partially suppressed by over-expression of the apoptosis inhibitors Diap1 or p35, which then results in small flight muscle fibers, emphasizing the essential role of Yorkie in promoting flight muscle growth.

How general is this role of Yorkie and the Hippo pathway in muscle fiber growth? Interestingly, muscle-specific loss (*Dlg5-IR*, *Slmap-IR* and *yorkie-IR*) and gain of function of Yorkie (*hippo-IR, yorkie-CA*) results in viable but flightless animals. This suggests that the larval muscles and the other adult *Drosophila* muscle fibers such as leg and abdominal muscles can form and function normally in the absence of Yorkie and the Hippo pathway. One reason might be their slower growth rates and their more limited sizes compared to indirect flight muscles. A second reason may relate to the particular myofibrillogenesis mechanism in flight muscles. Flight muscles display individual distinct myofibrils, which all form simultaneously at about 32 hr APF (*Weitkunat et al., 2014*) and then grow in length and diameter to match the volume increase of the muscle fiber (*Spletter et al., 2018*).

The closest mammalian homolog to insect flight muscles is the mammalian heart. Similar to flight muscles, the heart is a very stiff muscle using a stretch-modulated contraction mechanism (*Shiels and White, 2008*). After birth, mammalian cardiomyocytes stop dividing and all organ growth is achieved post-mitotically by size increase of the individual contractile cells. Interestingly, it was shown that mammalian YAP1 can promote cardiomyocyte survival and growth, as post-mitotic deletion of YAP1 results in increased fibrosis and cardiomyocyte apoptosis (*Del Re et al., 2013*). However, a role for the Hippo pathway in mammalian muscle is not limited to the heart. Constitutive expression of active YAP in adult mouse muscle fibers induces muscle fiber atrophy and deterioration of muscle function (*Judson et al., 2013*). Furthermore, it has been shown that the mammalian Hippo homolog Mst1 is a key regulator in fast skeletal muscle atrophy (*Wei et al., 2013*), and more importantly YAP promotes muscle fiber growth by its transcriptional activity requiring Tead cofactors (*Watt et al., 2015*). This suggests that the Hippo pathway via its transcriptional regulators Yorkie/YAP/TAZ is a general regulator of muscle fiber growth and survival in animals.

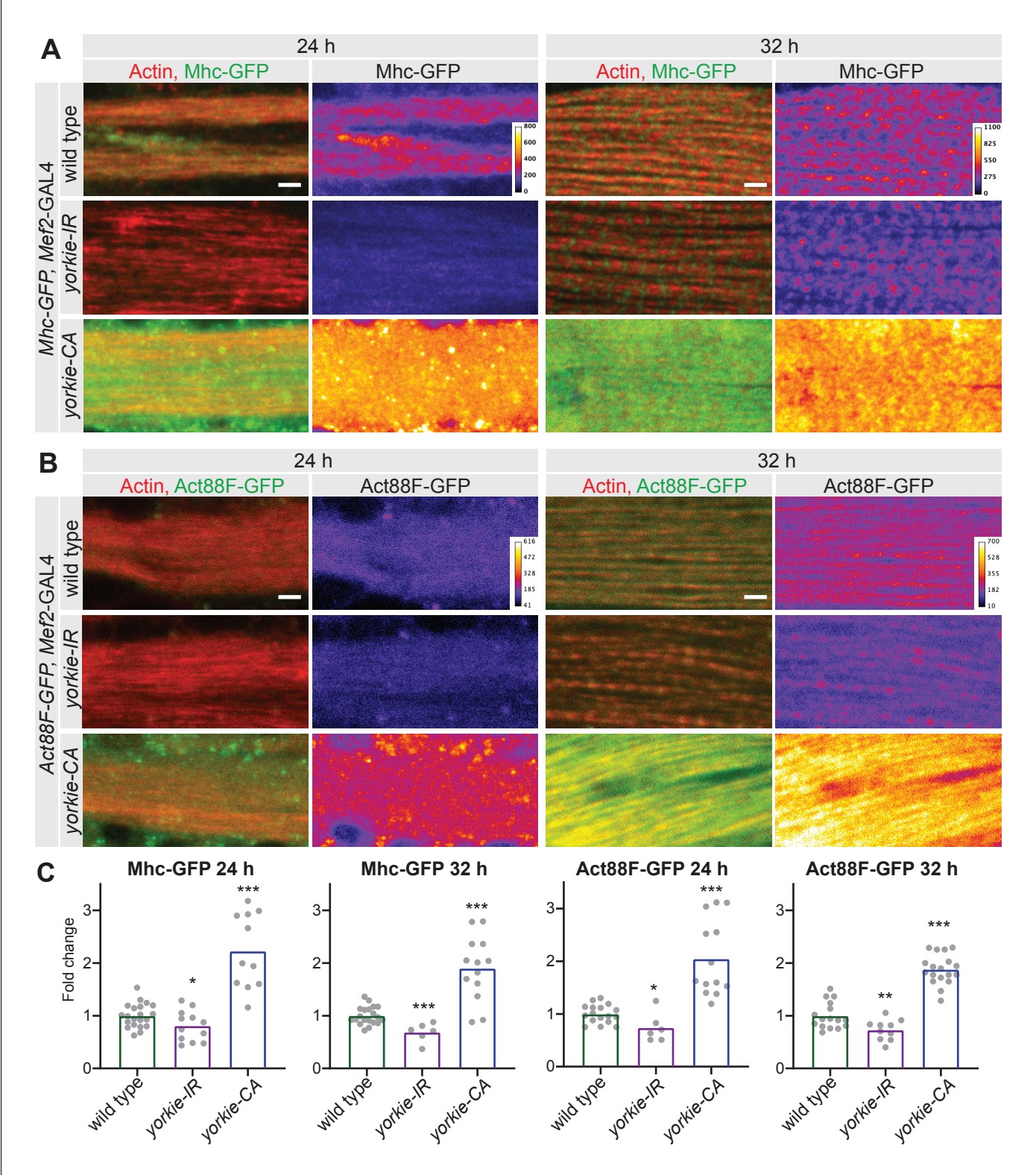

**Figure 9.** Yorkie regulates sarcomere protein expression. (**A, B**) Mhc-GFP (**A**) and Act88F-GFP (**B**) proteins levels visualised with a GFP nanobody together with actin comparing wild-type control with *yorkie-IR* and *yorkie-CA* at 24 hr and 32 hr after puparium formation (APF). Scale bars represent 2 μm. GFP channel is displayed using the 'fire' look-up table. (**C**) Quantification of relative GFP protein levels compared to wild type (wild-type mean set

*Figure 9 continued on next page*

Figure 9 continued

to one for each individual experiment). Bar plots display average normalised value and each dot represents one hemithorax. Student's t test, ***
p-value<0.001, ** p-value<0.01 * p-value<0.05.

The online version of this article includes the following figure supplement(s) for figure 9:

**Figure supplement 1.** Yorkie regulates sarcomere protein expression.

## Yorkie targets

How does Yorkie mediate flight muscle growth? Our transcriptomics analysis revealed that mRNAs coding for sarcomeric and mitochondrial components are major transcriptional targets of Yorkie in flight muscle. In flies and mammals, Yorkie/YAP/TAZ require transcriptional cofactors of the Tead family (TEA domain containing) to bind to DNA. Interestingly, it has been shown that mammalian homologs Tead1-4 bind to a DNA motif called 'muscle CAT' (MCAT), a motif well-known to regulate cardiac skeletal sarcomeric protein expression, including cardiac Troponin T or cardiac actin (*Carson et al., 1996*; *Farrance and Ordahl, 1996*; *Farrance et al., 1992*; *Wackerhage et al., 2014*). The only *Drosophila* Tead family member Scalloped binds to a very similar MCAT motif (*Wu et al., 2008*), and we found loss of *scalloped* shows the same phenotype as loss-of-function of *yorkie*. This strongly suggests that Yorkie and Scalloped cooperate in *Drosophila* muscle to transcriptionally boost sarcomeric gene expression to enable myofibril assembly and flight muscle fiber growth.

It has been shown that flight muscle fate is determined by the zinc finger transcriptional regulator Spalt major (*Schönbauer et al., 2011*). This includes the regulation of all flight muscle-specific sarcomeric components. How Yorkie cooperates with Spalt is to date an open question. One interesting link is that Yorkie and the Hippo pathway are required for the normal expression of *bruno1* (*bru1*, see *Figure 8B*). Bruno is the major splice regulator of flight muscle alternative splicing of sarcomeric proteins, downstream of Spalt (*Spletter et al., 2015*). Additionally, Bruno was shown to bind to 3'-UTRs of mRNA to regulate their translation efficiencies (*Webster et al., 1997*). Another RNA-binding protein that requires the Hippo pathway for normal expression is Imp (IGF-II mRNA-binding protein, see *Figure 8B*). Imp has been shown to regulate the stability and translation of a number of F-actin regulators (*Medioni et al., 2014*), suggesting that post-transcriptional effects of Hippo signalling can play an important role, too. This may explain the strong upregulation of Mhc, Act88F, Sls, and Prm proteins levels in *yorkie-CA* muscle at 24 hr APF despite little changes at the mRNA level. A similar crosstalk between the Hippo pathway and translational regulation via the mTORC1 complex has been suggested in mammals (*Hansen et al., 2015*).

## Regulation of the Hippo pathway

How is the Hippo pathway regulated to induce sarcomeric protein expression at the correct time during muscle development? Salvador, Expanded, Merlin, and Kibra, which regulate activity or plasma membrane localisation of Hippo in epithelial cells, all appear not to be required in flight muscles (*Schnorrer et al., 2010*; and data not shown). However, we find that the core members of the STRIPAK complex Slmap, Strip and Cka as well as Dlg5 are required to regulate Hippo in muscle. Interestingly, Strip and Cka have been shown to bind to each other and regulate either Hippo or other signalling pathways in other *Drosophila* tissues including eye, testis or motor neurons (*La Marca et al., 2019*; *Neal et al., 2020*; *Neisch et al., 2017*). Slmap appears to be the specific adaptor to link STRIPAK to Hippo in muscle (our work) and epithelia in flies (*Neal et al., 2020*; *Zheng et al., 2017*) as well as in mammalian cell culture (*Bae et al., 2017*; *Kwan et al., 2016*). As Slmap has a single transmembrane domain, it is likely localised to vesicles and our antibody stainings are consistent with at least a partial co-localisation to the assembling T-tubules, membrane invaginations that form during myofibrillogenesis and link the plasma membrane to the forming sarcomeres and the sarcoplasmic reticulum (*Peterson and Krasnow, 2015*; *Razzaq et al., 2001*; *Sauerwald et al., 2019*). Slmap may either bind or recruit Dlg5 to membranes, both were interestingly found in a complex with Mst1 (Hippo homolog) in mammalian cell culture (*Kwan et al., 2016*), which may boost the interaction of STRIPAK with Hippo, resulting in its effective de-phosphorylation. A detailed molecular model of Hippo signalling (*Figure 10*) remains speculative to date until the regulation of the precise physical interactions between the key components, Hippo, Slmap, Dlg5, and

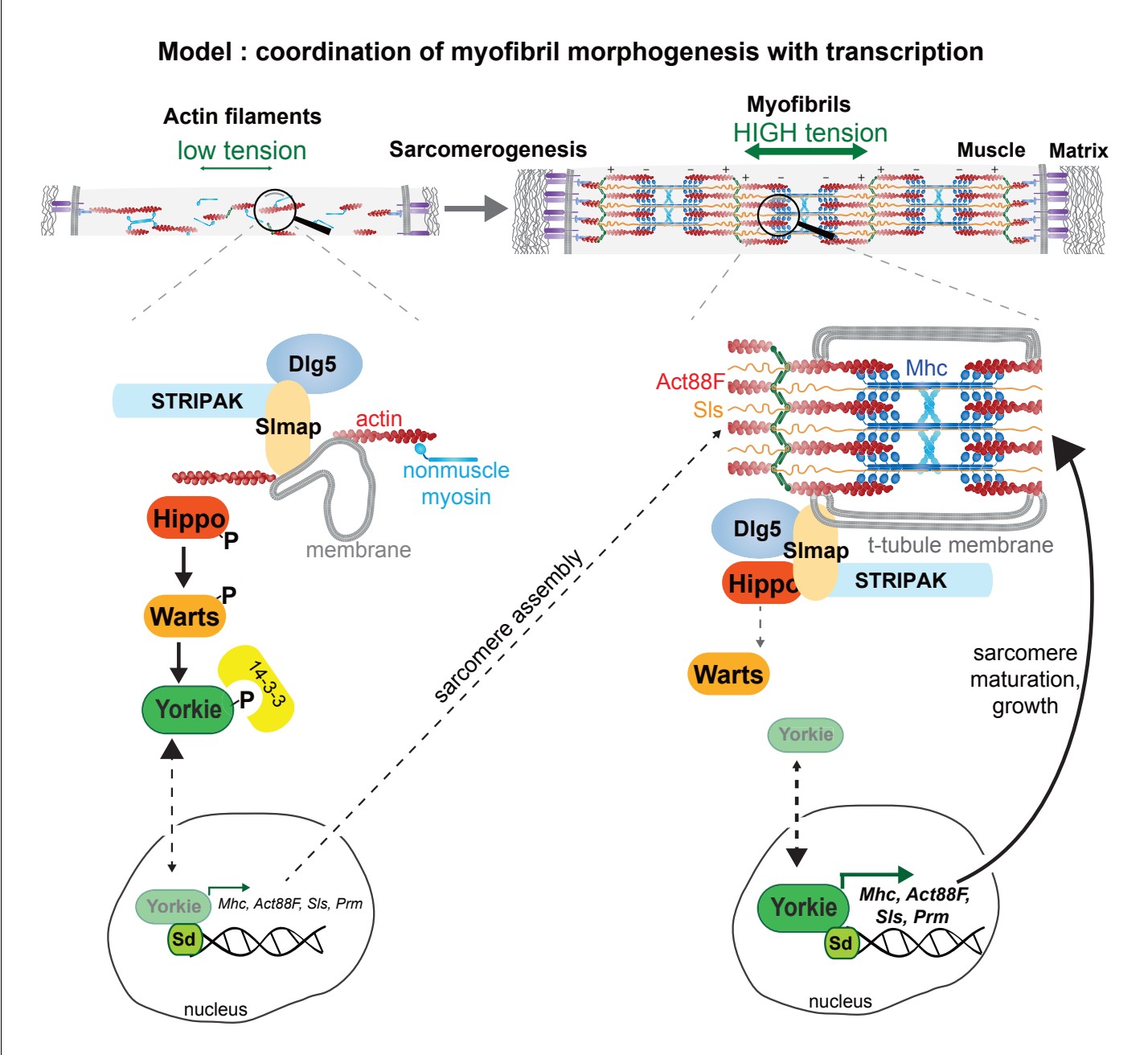

**Figure 10.** Model of Yorkie's role how to coordinate myofibrillogenesis with transcription during muscle morphogenesis.

Yorkie have been resolved in developing muscle. Interaction proteomics using dissected flight muscles at different stages may reveal interesting dynamics.

## A speculative model how myofibrillogenesis may coordinate with transcription

We have shown that mechanical tension in flight muscles increases during the early phases of myofiber development and identified it as a key regulator of myofibril assembly in flight muscle (*Weitkunat et al., 2014*). Concomitantly, transcription of sarcomeric components needs to be upregulated to allow myofibril assembly and myofibril maturation. An attractive although speculative model places the Hippo pathway central to coordinate the assembly status of the myofibrils with the transcriptional status of the muscle fiber nuclei (*Figure 10*). In analogy to the epithelial cell cortex,

we hypothesise that mechanical stretch produced by the actin cytoskeleton potentially via membranes, STRIPAK and Dlg5 may inhibit Hippo and thus promote the nuclear localisation of Yorkie, which in turn will boost transcription of mRNAs coding for sarcomeric components. This would be an analogous mechanism to the inhibition of Hippo by high tension produced by the cortical actomyosin and spectrin networks at the epithelial cortex resulting in epithelial tissue growth (*Deng et al., 2015*; *Fletcher et al., 2015*; *Fletcher et al., 2018*; *Rauskolb et al., 2014*). Interestingly, it was shown that filamentous actin (F-actin) levels can directly regulate Hippo pathway activity; more F-actin blocks Hippo signalling, resulting in epithelial tissue over-growth (*Fernández et al., 2011*; *Sansores-Garcia et al., 2011*; *Yin et al., 2013*). A similar feedback loop may enable the induction of more actin and myosin once the myofibrils have started to assemble and produce high levels of mechanical tension (*Figure 10*). This feedback model implies that only the successful completion of the myofibril assembly step then allows to fully boost sarcomeric mRNA transcription and translation to enable myofibril maturation and muscle fiber growth. This is consistent with the temporal separation of myofibril assembly and maturation in flight muscles (*González-Morales et al., 2019*; *Shwartz et al., 2016*; *Spletter et al., 2018*) and with the fiber growth defects found here when tension generation and hence myofibrillogenesis are compromised upon *kon* knock-down. Mechanical control of the Hippo pathway and YAP/TAZ localisation by the actomyosin cytoskeleton is also prominent in mammalian cells (*Dupont et al., 2011*; *Wada et al., 2011*). A similar mechanical feedback model as proposed here may therefore also be a relevant mechanism to coordinate post-mitotic cardiomyocyte and skeletal muscle growth or regeneration in mammals.

## Materials and methods

### Key resources table

| Reagent type (species) or resource | Designation | Source or reference | Identifiers | Additional information |
|---|---|---|---|---|
| Genetic reagent (*D. melanogaster*) | *w-* | BDSC: 3605 | | |
| Genetic reagent (*D. melanogaster*) | *Dlg5-IR-1* | VDRC: 22496 | | Pruned from the VDRC collection |
| Genetic reagent (*D. melanogaster*) | *Dlg5-IR-2* | VDRC: 101596 | | |
| Genetic reagent (*D. melanogaster*) | *Dlg5-IR-3* | BDSC: 30925 | w[*]; P{w[+mC]=UAS-Dlg5.RNAi}3, source: Helena Richardson | |
| Genetic reagent (*D. melanogaster*) | *Slmap-IR-1* | VDRC: 8199 | | |
| Genetic reagent (*D. melanogaster*) | *Slmap-IR-2* | BDSC: 32509 | y[1] sc[*] v[1] sev[21]; P{y[+t7.7] v[+t1.8]=TRiP.HMS00513}attP2 | |
| Genetic reagent (*D. melanogaster*) | *Cka-IR-1* | BDSC: 28927 | y[1] v[1]; P{y[+t7.7] v[+t1.8]=TRiP.HM05138}attP2 | |
| Genetic reagent (*D. melanogaster*) | *Cka-IR-2* | VDRC: 35234 | | |
| Genetic reagent (*D. melanogaster*) | *Strip-IR* | BDSC: 34657 | y[1] sc[*] v[1] sev[21]; P{y[+t7.7] v[+t1.8]=TRiP.HMS01134}attP2 | |
| Genetic reagent (*D. melanogaster*) | *yorkie-IR-1* | VDRC: 111001 | | |

*Continued on next page*

*Continued*

| Reagent type (species) or resource | Designation | Source or reference | Identifiers | Additional information |
|---|---|---|---|---|
| Genetic reagent (*D. melanogaster*) | *yorkie-IR-2* | VDRC: 40497 | | |
| Genetic reagent (*D. melanogaster*) | *warts-IR-1* | VDRC: 111002 | | |
| Genetic reagent (*D. melanogaster*) | *warts-IR-2* | BDSC: 41899 | y[1] sc[*] v[1] sev[21]; P{y[+t7.7] v[+t1.8]= TRiP.GL01331}attP2 | |
| Genetic reagent (*D. melanogaster*) | *Hippo-IR* | VDRC: 104169 | | |
| Genetic reagent (*D. melanogaster*) | *sd-IR* | VDRC: 101497 | | |
| Genetic reagent (*D. melanogaster*) | *kon-IR* | VDRC: 106680 PMID:24631244 | | |
| Genetic reagent (*D. melanogaster*) | *UAS-Dlg5-GFP* | BDSC: 30927 | w[*]; P{w[+mC]= UAS-Dlg5.RNAi}2, source: Helena Richardson | |
| Genetic reagent (*D. melanogaster*) | *Mef2-GAL4* | BDSC: 27390 | y[1] w[*]; P{w[+mC]=GAL4-Mef2.R}3 | |
| Genetic reagent (*D. melanogaster*) | *UAS-Flp; Mef2-GAL4* | this study | Mef2GAL4 on third combined with UAS-Flp on X | |
| Genetic reagent (*D. melanogaster*) | *UAS-GFP-Gma* | BDSC: 31776 PMID:11784030 | | |
| Genetic reagent (*D. melanogaster*) | *Tub-GAL80-ts* | BDSC: 7016 PMID:14657498 | | |
| Genetic reagent (*D. melanogaster*) | *hippo[wt]* | Duojia Pan, UT Soutwestern, USA PMID:29262338 | tub-FRT-y+-FRT-FLAG-hpo | |
| Genetic reagent (*D. melanogaster*) | *hippo[4A/431T]* | Duojia Pan, UT Soutwestern, USA PMID:29262338 | tub-FRT-y+-FRT-FLAG-hpo[4A/431T] | |
| Genetic reagent (*D. melanogaster*) | *Act88F-GAL4* | Richard Cripps, San Diego State University, USA PMID:22008792 | | |
| Genetic reagent (*D. melanogaster*) | *UAS-Hippo* | Barry Thompson, John Curtin School of Medical Research, Australia PMID:14502295 | | |
| Genetic reagent (*D. melanogaster*) | *UAS-Diap1* | BDSC: 6657; Barry Thompson, John Curtin School of Medical Research, Australia | w[*]; P{w[+mC]= UAS-DIAP1.H}3 | |
| Genetic reagent (*D. melanogaster*) | *UAS-p35* | BDSC:5072; Barry Thompson, John Curtin School of Medical Research, Australia | | |
| Genetic reagent (*D. melanogaster*) | *yorkie-CA* | BDSC: 28836; Barry Thompson, John Curtin School of Medical Research, Australia PMID:18256197 | y[1] w[*]; P{w[+mC]= UAS-yki.S168A. GFP.HA}10-12-1 | |

*Continued on next page*

*Continued*

| Reagent type (species) or resource | Designation | Source or reference | Identifiers | Additional information |
|---|---|---|---|---|
| Genetic reagent (*D. melanogaster*) | *UAS-Yki* | Richard G. Fehon, University of Chicago, USA PMID:30032991 | wild type Yorkie tagged with FLAG | |
| Genetic reagent (*D. melanogaster*) | *UAS-myr-Yki* | Richard G. Fehon, University of Chicago, USA PMID:30032991 | membrane tethered wild type Yorkie tagged with FLAG | |
| Genetic reagent (*D. melanogaster*) | *Mhc-GFP* | VDRC: 318471 PMID:26896675 | fTRG500 | |
| Genetic reagent (*D. melanogaster*) | *Act88F-GFP* | VDRC: 318362 PMID:26896675 | fTRG10028 | |
| Genetic reagent (*D. melanogaster*) | *Prm-GFP* | VDRC: 318114 PMID:26896675 | fTRG475 | |
| Genetic reagent (*D. melanogaster*) | *Sls-GFP [CA06744]* | Allan Spradling, Carnegie Institution for Science, Maryland, USA PMID:17194782 | | |
| Chemical compound, drug | GFP-Booster Atto488 | ChromoTek GmbH gba488 | | |
| Chemical compound, drug | Rhodamine Phalloidin | Thermo Fischer Scientific R415 | | |
| Chemical compound, drug | Vectashield with DAPI | Biozol VEC-H-1200 | | |
| Chemical compound, drug | Fluoroshield with DAPI | Sigma F6057 | | |
| Chemical compound, drug | anti-Dlg5, rabbit | this study | | |
| Chemical compound, drug | anti-Slmap, rabbit | this study | | |
| Chemical compound, drug | anti-Dlg1, mouse | DSHB: 4F3 PMID:11709153 | | |
| Chemical compound, drug | anti-Hippo, guinea pig | Georg Halder, VIB-KU Leuven, Belgium PMID:16341207 | | |
| Chemical compound, drug | anti-Amphiphysin, rabbit | Jörg Großhans, Georg-August-Universität, Göttingen, Germany PMID:11711432 | | |
| Chemical compound, drug | Tissue-Tek O.C.T. | Weckert Labotechnik 4583; Sakura Finetek | | |
| Chemical compound, drug | FEATHER Microtome Blades, C35 | pfmmedical, Cologne, Germany 207500003 | | |
| Chemical compound, drug | TRIzol | Thermo Fischer Scientific 15596026 | | |

## Fly strains and genetics

Fly strains were maintained using standard conditions. Unless otherwise stated, all experiments were performed at 27°C to improve RNAi efficiency. When applying temperature-sensitive *Tub*-GAL80ts, fly crosses were kept at 18°C to suppress GAL4 activity and the white pre-pupae (0–30 min APF) were shifted to 31°C to allow GAL4 activity only at pupal stages. The pupae were then raised at 31°C

until the desired age. Considering that pupae develop faster at 31°C compared to 27°C, timing was corrected by growing pupae at 31°C for 30 hr age matching 32 hr APF at 27°C, for 44 hr matching 48 hr APF and for 84 hr matching 90 hr APF at 27°C. The *Act88F*-GAL4 *UAS-Hippo* pupae were raised for 66 hr at 18°C age matching 32 hr APF at 27°C, and for 96 hr at 18°C age matching 48 hr APF at 27°C. RNAi stocks used were from the Vienna (*Dietzl et al., 2007*) or Harvard collections (*Ni et al., 2011*) and obtained from VDRC or Bloomington stock centers. All used fly strains are listed in Key Resource Table.

## Flight

Flight tests were performed as previously described (*Schnorrer et al., 2010*). One- to three-day-old male flies were dropped into a 1 m long plexiglass cylinder with 8 cm diameter and five marked sections. Wild-type flies land in the upper two sections, whereas flies with defective flight muscles fall to the bottom of the tube immediately. For each genotype, flight assays were performed two or three times with minimum seven males each (for *yorkie-IR* females were used) (*Supplementary file 1*).

## Pupal dissection and flight muscle stainings

For 24 hr, 32 hr, and 48 hr APF pupal dissections, the pupa was stabilised on a slide by sticking to double-sticky tape. The pupal case was removed with fine forceps. Using insect pins two or three-holes were made in the abdomen to allow penetration of the fixative and pupae were fixed in 4% PFA (Paraformaldehyde) in PBST (PBS with 0.3% Triton-X) in black embryo glass dishes for 15 min at room temperature (RT). After one wash with PBST, the pupae were immobilised using insect pins in a silicone dish filled with PBST and dissected similarly as described previously (*Weitkunat and Schnorrer, 2014*). Using fine scissors, the ventral part of each pupa was removed, then the thorax was cut sagittally and the thorax halves were freed from the abdomen, leaving half thoraces with the flight muscles on them. These half thoraces were transferred to black embryo dishes and blocked for 30 min at room temperature (1/30 normal goat serum in PBST). Primary antibodies (anti-Amph: rabbit, 1/500 (*Razzaq et al., 2001*); anti-Dlg1: mouse, 1/500, DSHB; anti-Hippo: guinea pig, 1/1000 (*Hamaratoglu et al., 2006*); anti-Dlg5: rabbit, animal 2125, 1/1000, pre-absorbed; anti-Slmap: rabbit, animal 2151, 1/1000, pre-absorbed) were incubated either at room temperature for 2 hr or overnight at 4°C. Half thoraces were then washed three times 10 min in PBST at room temperature and incubated with secondary antibodies coupled to Alexa dyes (Alexa488 or Alexa568, all from Molecular Probes, 1/500) for 2 hr at room temperature in the dark. To visualise F-actin rhodamine-phalloidin was added to the secondary antibody solution (Molecular Probes, 1/500 in PBST). To visualise GFP fusion proteins, GFP-booster conjugated with Atto488 (ChromoTek, 1/200 in PBST) in combination with rhodamine-phalloidin was used. Half thoraces were incubated 2 hr at room temperature. After three washes with PBST, half thoraces were mounted using Vectashield including DAPI (Biozol).

For comparing protein levels in wild type versus knock-down conditions, pupae from wild type and knock-down genotypes were dissected, stained, and imaged on the same day in parallel under identical settings (master mixes for staining reagents, identical laser and scanner settings for imaging).

For 90 hr APF pupal dissections, the head, wings, legs, and abdomen were cut off the thorax with fine scissors, and the thoraxes were fixed for 20 min in 4% PFA in PBST at RT. After washing once with PBST, the thoraxes were placed on a slide with double-sticky tape and cut sagittally (dorsal to ventral) with a microtome blade (Pfm Medical Feather C35). These half thoraces were stained similarly to the early pupa half thoraces and mounted in Vectashield with DAPI using two spacer coverslips on each side.

## Production of polyclonal antibodies against Dlg5 and Slmap

*Dlg5* cDNA (Gold clone LD32687) corresponding to amino acids 74–379 and *Slmap* cDNA (Gold clone LD47843) for the full protein sequence were PCR amplified, cloned into pCoofy4 (*Scholz et al., 2013*), expressed in *E. coli* to generate His6-MBP fusion proteins; the tags were cleaved and peptides were injected into rabbits using standard protocols (Max Planck Institute of Biochemistry service facilities).

## Image acquisition and processing

Image acquisition was performed with Zeiss LSM780-I-NLO, LSM880-I-NLO and LSM880-U-NLO confocal microscopes using Plan Neofluar 10x/0.30 NA air, Plan-Apo 40x/1.4 NA oil and Plan-Apo 63x/1.4 NA oil objectives. For all samples z-stacks were acquired. Image processing was done using Fiji (*Schindelin et al., 2012*). Digital cross-sections were created from z-stacks that covered the entire width of the flight muscle DLM4 by drawing a straight line in the central part of the fiber and re-slicing. For displaying the thick digital cross-sections, 'Bleach correction' plug-in of Fiji was used with Histogram Matching Method. Fiber length and fiber cross-sectional area were measured with freehand drawing tools in Fiji based on phalloidin staining. To determine the average muscle fiber length per hemithorax, the length of all flight muscles for which both fiber ends were visible were measured and averaged.

## Myofibril length quantification and intensity profile plots

Myofibrils stained with phalloidin from a 40 µm x 20 µm x 2.5 µm confocal microscope stack were traced semi-automatically using Simple Neurite Tracer plug-in in Fiji (*Longair et al., 2011*). In each pupa, 10 myofibrils were traced and their average length was calculated. To visualise sarcomere periodicity, intensity profiles were plotted along a line based on actin labelling (phalloidin) using Fiji.

## Nuclei count

To count nuclei numbers of flight muscle fibers, half thoraces were stained with phalloidin (actin) and DAPI (nuclei) and imaged first using 10x objective to quantify the entire length of the fiber and then with 63x oil objective to visualise details. The acquired 63x z-stacks contain the entire muscle depth and about half of the muscle fiber length. Using Fiji's multi-point tool all the nuclei in each z-stack were counted manually, using actin labelling as a landmark to visualise the borders of the fiber. Nuclei number for the entire fiber was calculated using the length of the entire fiber from the 10x image.

## Cryo cross-sections

Cryo cross-sections were performed as described previously (*Spletter et al., 2018*). Briefly, the pupal case was removed by fine forceps. Using insect pins, two or three holes were made in the abdomen to allow penetration of the 4% PFA in PBST (PBS with 0.5% Triton-X) overnight at 4°C. Fixed pupae were sunk in 30% sucrose solution in PBST on a nutator at 4°C. Pupae were embedded in O.C.T. compound in plastic moulds (#4566, Sakura Finetek) and frozen on dry ice. Blocks were sectioned at 20 µm thickness on a microtome cryostat. Sections were adhered on glass slides coated with 1% gelatin, 0.44 mM chromium potassium sulfate dodecahydrate to improve tissue adherence. Sections on the slide were fixed for 1 min in 4% PFA with PBS at room temperature, washed once for 5 min in PBS, incubated with rhodamine-phalloidin (1/500 in PBST) for 2 hr at RT in a wet chamber, washed three times with PBST and mounted in Fluoroshield with DAPI.

## Quantifying GFP protein levels

GFP-tagged genomic fosmid fly lines (fTRG500 for Mhc-GFP, fTRG10028 for Act88F-GFP, fTRG475 for Prm-GFP) (*Sarov et al., 2016*) and GFP-trap fly line Sls-GFP [CA06744] (*Buszczak et al., 2007*) were used for comparison of protein levels in *Mef2*-GAL4 driven knockdown of *yorkie* or gain-of-function *yorkie-CA* to *Mef2*-GAL4 control. Pupae of the different genotypes were dissected, stained, and imaged on the same day in parallel under identical settings (master mixes for staining reagents, identical laser and scanner settings for imaging). Per hemi-thorax one or two different areas were imaged using 63x oil objective, zoom factor 2. In Fiji, five z planes at comparable positions in the muscles were selected for an average intensity projection of the volume into a 2D plane. In the 2D plane, two or three regions of 50 µm$^2$ occupied by myofibrils (based on actin labelling) were selected. Mean intensities of each of these regions were averaged to calculate one value per hemi-thorax used for the quantification graphs. For each experimental day, the mean intensity of all wild-type samples was set to one to calculate the relative intensity of the other genotypes. Data from a minimum of two independent experiments were plotted.

## RNA-isolation from developing flight muscle

For each replicate, flight muscles from seven *Mef2*-GAL4, *UAS-GFP-Gma* pupae at 24 hr or 32 hr APF were dissected in ice-cold PBS treated with DEPC using a fluorescent binocular. Flight muscles were collected in an Eppendorf tube and centrifuged at 2000 g for 5 min. The flight muscle pellet was re-suspended in TRIzol, shock-frozen in liquid nitrogen and kept at −80°C.

RNA was isolated directly from the TRIzol muscle samples using a 96-well plate extraction kit (Direct-zol−96 RNA, Zymo Research, #R2054): after thawing to room temperature in 1.5 ml Eppendorf tubes, the tissue samples were homogenised using a small pestle, followed by nucleic acid precipitation with 100% ethanol. This suspension was then transferred to the 96-well plate containing the purification columns. DNA digestion was performed 'in column' according to the kit instructions. Total RNA was eluted with 25 µl of RNase-free water and was quantified using the Quantifluor RNA System (Promega, #E3310).

## BRB-sequencing library preparation

RNA sequencing libraries were prepared using 20 ng of total RNA following the BRB-sequencing protocol (*Alpern et al., 2019*). Briefly, each RNA sample was reverse transcribed in a 96-well plate using SuperScriptTM II Reverse Transcriptase (Lifetech 18064014) with individual barcoded oligo-dT primers (Microsynth, Switzerland. For primer sequences see *Alpern et al., 2019*). Next, the samples were split into three pools, purified using the DNA Clean and Concentrator kit (Zymo Research #D4014), and treated with exonuclease I (New England BioLabs, NEB #M0293S). Double-stranded cDNA was generated by the second strand synthesis via the nick translation method. For that, a mix containing 2 µl of RNAse H (NEB, #M0297S), 1 µl of *E. coli* DNA ligase (NEB, #M0205 L), 5 µl of *E. coli* DNA Polymerase (NEB, #M0209 L), 1 µl of dNTP (0 .2 mM), 10 µl of 5x Second Strand Buffer (100 mM Tris, pH 6.9, AppliChem, #A3452); 25 mM $MgCl_2$ (Sigma, #M2670); 450 mM KCl (Appli-Chem, #A2939); 0.8 mM (β-NAD Sigma, N1511); 60 mM $(NH_4)_2SO_4$ (Fisher Scientific Acros, #AC20587); and 11 µl of water was added to 20 µl of ExoI-treated first-strand reaction on ice. The reaction was incubated at 16°C for 2.5 hr. Full-length double-stranded cDNA was purified with 30 µl (0.6x) of AMPure XP magnetic beads (Beckman Coulter, #A63881) and eluted in 20 µl of water.

The Illumina compatible libraries were prepared by tagmentation of 5 ng of full-length double-stranded cDNA with 1 µl of in-house produced Tn5 enzyme (11 µM). After tagmentation the libraries were purified with DNA Clean and Concentrator kit (Zymo Research #D4014), eluted in 20 µl of water and PCR amplified using 25 µl NEB Next High-Fidelity 2x PCR Master Mix (NEB, #M0541 L), 2.5 µl of P5_BRB primer (5 µM, Microsynth), and 2.5 µl of Illumina index adapter (Idx7N5 5 µM, IDT) following program: incubation 72°C—3 min, denaturation 98°C—30 s; 15 cycles: 98°C—10 s, 63°C—30 s, 72°C—30 s; final elongation at 72°C—5 min. The fragments ranging 200–1000 bp were size-selected using AMPure beads (Beckman Coulter, #A63881) (first round 0.5x beads, second 0.7x). The libraries were profiled with High Sensitivity NGS Fragment Analysis Kit (Advanced Analytical, #DNF-474) and measured with Qubit dsDNA HS Assay Kit (Invitrogen, #Q32851) prior to pooling and sequencing using the Illumina NextSeq 500 platform using a custom primer and the High Output v2 kit (75 cycles) (Illumina, #FC-404–2005). The library loading concentration was 2.2 pM and sequencing configuration as following: R1 6 c / index 8 c / R2 78 c.

## Pre-processing of the data—de-multiplexing and alignment

The sample reads de-multiplexing was done using BRB-seqTools (http://github.com/DeplanckeLab/BRB-seqTools) as described before (*Alpern et al., 2019*). The sequencing reads were aligned to the Ensembl gene annotation of the *Drosophila melanogaster* BDGP6.23 genome using STAR (version 020201) (*Dobin et al., 2013*), and count matrices were generated with HTSeq (version 0.9.1) (*Love et al., 2014*).

## Bioinformatics analysis of BRB-Seq data

BRB-seq data quality was assessed in several ways. First, we excluded five samples from further analysis due to low numbers of aligned reads (<500K; removed were 1 × 24 hr APF wild type, 1x *Dlg5-IR* 24 hr, 1x *yorkie-CA* 24 hr and 2 × 32 hr APF wild type). Using raw read counts, we performed PCA analysis, calculated heatmaps and Pearson's correlation in R (Version 3.3.1, https://cran.r-project.org/). One additional replicate from 32 hr APF wild type was removed from further analysis as it

represented a clear outlier. For the remaining samples, we performed library normalisation (RLE) as well as differential expression analysis using DESeq2 (version 1.22.2, *Love et al., 2014*). Genes were considered differentially expressed with a log2FC $\geq$ |1| and an FDR $\leq$ 0.05. Functional enrichment analysis was performed on the differentially expressed genes using FlyEnrichr (*Kuleshov et al., 2019*). Data are available at the Gene Expression Omnibus database (*Clough and Barrett, 2016*), accession number GSE158957.

### Affinity enrichment mass spectrometry

For each replicate, about 100 pupae staged from 24 hr - 48 hr APF of the genotypes *Mef2*-GAL4 (control) or *Mef2*-GAL4, *UAS-Dlg5-GFP* were collected and processed as described previously (*Sarov et al., 2016*). For each genotype, four replicates (100 pupae each) were snap-frozen in liquid nitrogen and ground to powder while frozen. The powder was processed as described (*Hubner et al., 2010*). Briefly, the cleared lysate was mixed with magnetic beads pre-coupled to a GFP antibody matrix to perform single step affinity enrichment and mass-spec analysis using an Orbitrap mass spectrometer (Thermo Fisher) and the QUBIC protocol (*Hein et al., 2015*). Raw data were analysed in MaxQuant version 1.4.3.22 (*Cox and Mann, 2008*) using the MaxLFQ algorithm for label-free quantification (*Cox et al., 2014*). The volcano plot was generated with Graphpad.

## Acknowledgements

The authors are indebted to Reinhard Fässler for hosting part of this work in his department, and to Jürg Müller and Anne-Kathrin Classen for hosting AK-Ç during parts of this study. We are grateful to the IBDM and LIC (Albert-Ludwigs University of Freiburg) imaging facilities for help with image acquisition and maintenance of the microscopes and the Max Planck service facilities for protein expression and antibody production. We acknowledge the France-BioImaging infrastructure supported by the French National Research Agency (ANR–10–INBS-04–01, Investments for the future). Fly stocks obtained from the Bloomington *Drosophila* Stock Center (NIH P40OD018537) and the Vienna *Drosophila* Resource Center (VDRC) were used in this study. The authors thank Barry Thompson, Richard Fehon, Jörg Großhans, Georg Halder and Duojia Pan for fly stocks and antibodies, as well as Pierre Mangeol and Clara Sidor for helpful discussions and critical comments for this manuscript.

## Additional information

### Funding

| Funder | Grant reference number | Author |
|---|---|---|
| European Research Council | FP/2007-2013 | Frank Schnorrer |
| Centre National de la Recherche Scientifique | | Frank Schnorrer |
| Aix-Marseille Université | ANR-11-IDEX-0001-02 | Frank Schnorrer |
| Agence Nationale de la Recherche | ANR-ACHN MUSCLE-FORCES | Frank Schnorrer |
| Agence Nationale de la Recherche | ANR-18-CE45-0016-01 | Bianca H Habermann |
| Human Frontier Science Program | RGP0052/2018 | Frank Schnorrer |
| Agence Nationale de la Recherche | ANR-10-INBS-04-01 | Frank Schnorrer |
| Humboldt Foundation | | Aynur Kaya-Çopur |
| EMBO | | Aynur Kaya-Çopur |
| Fondation Bettencourt Schueller | | Frank Schnorrer |
| Turing Center for Living Sys- | | Frank Schnorrer |

| | | |
|---|---|---|
| tems | | |
| Max Planck Society | Frank Schnorrer | |

The funders had no role in study design, data collection and interpretation, or the decision to submit the work for publication.

### Author contributions

Aynur Kaya-Çopur, Conceptualization, Data curation, Formal analysis, Investigation, Methodology, Writing - original draft, Writing - review and editing; Fabio Marchiano, Formal analysis, Methodology, Writing - original draft; Marco Y Hein, Daniel Alpern, Data curation, Formal analysis, Methodology; Julie Russeil, Data curation, Methodology; Nuno Miguel Luis, Data curation, Methodology, Writing - review and editing; Matthias Mann, Bart Deplancke, Bianca H Habermann, Resources, Supervision, Methodology; Frank Schnorrer, Conceptualization, Resources, Supervision, Funding acquisition, Writing - original draft, Writing - review and editing

### Author ORCIDs

Aynur Kaya-Çopur ⓘD https://orcid.org/0000-0002-3161-9442
Marco Y Hein ⓘD http://orcid.org/0000-0002-9490-2261
Nuno Miguel Luis ⓘD http://orcid.org/0000-0001-5438-9638
Matthias Mann ⓘD http://orcid.org/0000-0003-1292-4799
Bart Deplancke ⓘD http://orcid.org/0000-0001-9935-843X
Bianca H Habermann ⓘD http://orcid.org/0000-0002-2457-7504
Frank Schnorrer ⓘD https://orcid.org/0000-0002-9518-7263

### Decision letter and Author response

Decision letter https://doi.org/10.7554/eLife.63726.sa1
Author response https://doi.org/10.7554/eLife.63726.sa2

# Additional files

### Supplementary files

• Supplementary file 1. Data table containing data from *Figures 1*, *3*, *4*, *5*, *6*, *7* and *9*.

• Supplementary file 2. Table listing the expression levels and fold changes as well as normalised p-values of the BRB-seq data compared to the wild-type controls. All or only the significantly different genes are listed.

• Supplementary file 3. GO enrichment terms of the significantly different gene lists of the various genotypes and time points.

• Transparent reporting form

### Data availability

Sequencing data have been deposited in GEO under accession code GSE158957.

The following dataset was generated:

| Author(s) | Year | Dataset title | Dataset URL | Database and Identifier |
|---|---|---|---|---|
| Kaya-Çopur A, Marchiano F, Hein MY, Alpern D, Russeil J, Luis NM, Mann M, Deplancke B, Habermann BH, Schnorrer F | 2020 | The Hippo pathway controls myofibril assembly and muscle fiber growth by regulating sarcomeric gene expression | https://www.ncbi.nlm.nih.gov/geo/query/acc.cgi?acc=GSE158957 | NCBI Gene Expression Omnibus, GSE158957 |

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
