## [Decision Letter]

**Acceptance summary:**

This is a valuable paper that links sarcomeric growth to sarcomeric transcription. The authors very nice link muscle growth that of growth the contractile apparatus using the flight muscles of the fruitfly *Drosophila*. They show that members of the Hippo pathway are needed for flight muscle growth and that the pathway controls sarcomeric gene expression.

**Decision letter after peer review:**

Thank you for submitting your article "The Hippo pathway controls myofibril assembly and muscle fiber growth by regulating sarcomeric gene expression" for consideration by *eLife*. Your article has been reviewed by three peer reviewers, and the evaluation has been overseen by a Reviewing Editor and K VijayRaghavan as the Senior Editor. The reviewers have opted to remain anonymous.

The reviewers have discussed the reviews with one another and the Reviewing Editor has drafted this decision to help you prepare a revised submission.

Summary

This is a deeply impressive manuscript. The authors identify a novel and unexpected role for the Hippo pathway in muscle development in *Drosophila*. A wide spectrum of genetic experiments demonstrates a fundamental role for Hippo signalling in regulating sarcomeric gene expression to control muscle growth. Of particular interest, the upstream regulation of Hippo signalling differs in muscle from what is seen in epithelial cells (where it depends on apical proteins). Instead, muscle cells use Dlg5 and SLMAP – key STRIPAK components – that directly control Hippo kinase activity in muscle. An insightful model is presented that draws together their findings and emphasises the importance of mechanical tension and the establishment of t-tubules as being crucial upstream events that may determine Dlg5-SLMAP activity during muscle maturation.

Overall, the results are novel, robustly demonstrated, and of major importance and significance to the fields of developmental biology and Hippo signalling.

Essential revisions

1) The manuscript by Kaya-Copur et al. focuses on the role of Hippo pathway during development of a subset of Indirect Flight Muscles, the Dorsal Longitudinal Muscles (DLMs). This is an extensive, well executed and nicely documented work linking phenotyping, transcriptomic and proteomic analyses and demonstrating that all actors and interactors of Hippo pathway play a role in compaction of developing DLMs, which takes place between 24 and 32h APF. Non compaction observed for example in Yorki RNAi context at 32h APF leads to affected myofibrils assembly, insufficient activation of sarcomeric genes and in muscle cells apoptosis with total DLM loss at 48h APF.

Interestingly, blocking apoptosis by overexpressing DIAP1 in Yorki RNAi context allows to maintain DLMs until 48h APF. However, they appear much thinner indicating that Hippo pathway, during DLM development, also plays its well-known role of cell/tissue size regulator.

Overall, it is a highly valuable work, linking requirement for tension during DLM compaction with the capacity to grow and increase in size by stimulating sarcomeric gene expression. The hypothetical model presented in the last figure appears as an attractive possibility.

2) My major concern is the evidence for Hippo as a regulator of muscle growth in *Drosophila*. This seems to be the case when DIAP1 rescues apoptosis in Dlg5RNAi or in YorkiRNAi contexts and we see smaller muscles at 48h APF. However, rescue of apoptosis appears temporal with subsequent loss of DLMs and thus this raises question of whether smaller muscles are truly effects of Hippo function or rather due to partial rescue of apoptosis, which could potentially affect muscle growth.

3) Because authors do not see effects of Hippo on the growth of other adult muscles such as leg muscles (as mentioned in discussion) an additional evidence for the role of Hippo pathway in the DLM growth is required. One possibility would be to see whether constitutively active Yorki expressed after compaction event could impact on (increase) the DLM size.

4) In this study the authors suggest that native Hippo signaling in the developing *Drosophila* DLMs controls sarcomere assembly. They arrive at the model outlined in Figure 9 by perturbing various known components of the Hippo pathway and interactors they gleaned from DLG5 targeted proteomics. The quality of data and extent of investigation are truly remarkable. In the following I bring certain points to the authors' attention to raise this manuscript beyond reproach.

5) Clearly most the manipulations show drastic sarcomere assembly, fiber length and fiber formation phenotypes in DLM. Inferences from these phenotypes rely on the assumption that Hippo pathway proteins and others in this study are expressed in developing DLMs. Similar assumptions were made in Yatsenko et al. BMC Medicine https://doi.org/10.1186/s12916-019-1478-3 and Yadav, Puah and Wasser, Royal Society Open Biology 2016. This notion is mostly supported by consistent RNAi phenotypes.

6) To a skeptic, the drastic phenotypes seen may be an artefact of misexpression that may be irrelevant to native cellular processes i.e. these proteins are usually not expressed in DLMs but disturb development if introduced. Therefore, the epistasis analysis of these misexpression phenotypes and their transcriptional consequences, i.e. co misexpression experiments in Figure 1 to Figure 8, may not inform us of the native function of the proteins in DLM development. These results may reflect non-native protein function in a developmentally anachronistic tissue context. Heavy reliance on RNAi without ascertaining off-target affects in DLMs at the target stage of development (true for previous publications as well), further confounds inferences from these results. It may be essential to address such concerns. This point can simply be addressed in the Discussion.

7) The authors use mass spec investigations to elucidate protein-protein interactions to support their interpretations, specifically Dlg5GFP driven in muscles under the *mef2*-Gal4 promoter. While these interactions may happen in other muscles, data need to unequivocally show that they occur in developing DLMs at the stages being investigated. Since the source tissue for these experiments is whole pupae, and DLMs at these stages may not out-represent interactions in other muscles through this measurement, DLM specific comments invite skepticism. The Schnorrer lab has previously shown that molecular regulation of DLM growth differs from other *Drosophila* muscle. Therefore, examination of DLG5 protein-protein interactions in developing DLMs, on which subsequent connections with Hippo signaling have been based, would have been useful. Here again, no experiments are required, but this point can be addressed in the Discussion.

8) Further, Dlg5 is a marker of T-Tubules (Peterson and Krasnow, 2015). Even if the proteomics were limited to DLMs, the authors would have to untangle the catastrophic effects of Dlg5 perturbation (Figure 1) in relation to Hippo signaling from oxygen availability to the developing muscle. The authors should address such a concern.

9) To ensure that their inferences are valid for DLMs specifically, the authors must first establish Hippo pathway protein expression and function. It is crucial that authors ascertain native expression of the Hippo, Yorkie and other member proteins of this signaling pathways specifically from isolated DLMs at the stages being investigated. Whole thoraces/pupae will not speak to DLM specific molecular processes. This can be achieved through IHCs. Visualizing Yki is notoriously difficult. Clean western blots from tagged transgenic lines showing normal development will suffice. The same procedures must be followed for known reporters of Yki function without overexpression. Further, if the expression of these proteins in the context of pupal DLMs can be demonstrated, the authors must show that these are in fact perturbed by manipulations. If these are not tenable experimentally, the authors should discuss the issue of the phenotypes upon driving expression, using the *mef2*-Gal4 driver, of Yki, YkiCA, Yki-myr, Hippo and Diap1, but not Dlg5; when we have no evidence for their expression in the muscles. Could this be an ectopic gain-of-function phenotype? If the authors can show expression data if they have them that will be nice, or else they should ratchet-down their interpretation appropriately. (The authors could show native expression and activity of Yki in developing DLMs, through IHC or WB through tagged Yki expression if necessary. Reporters like diap1-GFP and ex-LacZ among others are also useful. Western blots or mass spectrometry from isolated developing DLMs of Yki, p-Yki and other Hippo pathway members are acceptable as well. If the effects of genetic manipulations on the native levels and function of these proteins can be shown, that will be very good. Else, please discuss potential concerns).

---

## [Author Response]

Essential revisions1) The manuscript by Kaya-Copur et al. focuses on the role of Hippo pathway during development of a subset of Indirect Flight Muscles, the Dorsal Longitudinal Muscles (DLMs). This is an extensive, well executed and nicely documented work linking phenotyping, transcriptomic and proteomic analyses and demonstrating that all actors and interactors of Hippo pathway play a role in compaction of developing DLMs, which takes place between 24 and 32h APF. Non compaction observed for example in Yorki RNAi context at 32h APF leads to affected myofibrils assembly, insufficient activation of sarcomeric genes and in muscle cells apoptosis with total DLM loss at 48h APF.Interestingly, blocking apoptosis by overexpressing DIAP1 in Yorki RNAi context allows to maintain DLMs until 48h APF. However, they appear much thinner indicating that Hippo pathway, during DLM development, also plays its well-known role of cell/tissue size regulator.Overall, it is a highly valuable work, linking requirement for tension during DLM compaction with the capacity to grow and increase in size by stimulating sarcomeric gene expression. The hypothetical model presented in the last figure appears as an attractive possibility.

We thank the reviewer for this very positive evaluation of our manuscript and in particular for appreciating the role of mechanical tension as important growth regulator in muscle, which we highlight in our model.

2) My major concern is the evidence for Hippo as a regulator of muscle growth in *Drosophila*. This seems to be the case when DIAP1 rescues apoptosis in Dlg5RNAi or in YorkiRNAi contexts and we see smaller muscles at 48h APF. However, rescue of apoptosis appears temporal with subsequent loss of DLMs and thus this raises question of whether smaller muscles are truly effects of Hippo function or rather due to partial rescue of apoptosis, which could potentially affect muscle growth.

It is indeed well documented that apoptosis and cell growth need to be coordinated to achieve tissue growth and the Hippo pathway controls both in most systems (see Udan et al., 2003 or Harvey and Tapon, 2007) as we also discuss in our paper. Possibly, we are not able to rescue apoptosis completely by over-expression of *Diap1* in the *Dlg5*and *yorkie* RNAi background. However, at 48h APF all muscles are present and they are very small (Figure 6A and B). The loss of some muscle fibers after 48 h APF in the rescued genotype may also be caused by mechanical ruptures that occur when muscles cannot match the growth of tendons and the cuticle. Thus, together with our transcriptomics data, we believe it is solid to conclude that Yorkie and Dlg5 are required for normal muscle growth.

To further substantiate this conclusion, we have extended our experimental evidence and blocked apoptosis in the *Dlg5* knock-down background with an even more potent inhibitor of apoptosis in many cellular contexts, by over-expressing baculovirus caspase inhibitor *UAS-*p35 (Clem et al., 1991). We find that over-expression of p35 can indeed rescue apoptosis of *Dlg5* knock-down muscles, which then also results in much smaller muscle fibers at 48h APF. Similar to the rescue with Diap1, some of these small fibers survive until 90 h APF (new Figure 6—figure supplement 1B, C). Together, these data strongly support our conclusion that the Hippo pathway is important for muscle growth.

3) Because authors do not see effects of Hippo on the growth of other adult muscles such as leg muscles (as mentioned in discussion) an additional evidence for the role of Hippo pathway in the DLM growth is required. One possibility would be to see whether constitutively active Yorki expressed after compaction event could impact on (increase) the DLM size.

It is demanding to ask for more muscle growth in the gain of function condition. We indeed do see a slight over-growth in *yorkie-CA* muscles at 24 h APF (Figure 5A, B), when driven with *Mef2*-GAL4. And we do show that *yorkie-CA* muscles display a strong gain of sarcomeric RNA and sarcomere protein expression at 32 h APF, to various degrees depending on the gene. We now provide additional protein quantification data for two more core sarcomeric components, Paramyosin (Prm) and the titin homolog Sallimus (Sls), both of which are strongly increased in *yorkie-CA* (new Figure 9—figure supplement 1). However, as the stoichiometry of the various sarcomeric components is likely not correct for sarcomere assembly in *yorkie-CA*, it is unclear if more myofibrils are assembled and if these muscles truly over grow. The fibers do display a defective shape caused by defective myofibril assembly and hence are difficult to quantify in volume, especially in later time points (Figure 5C).

4) In this study the authors suggest that native Hippo signaling in the developing *Drosophila* DLMs controls sarcomere assembly. They arrive at the model outlined in Figure 9 by perturbing various known components of the Hippo pathway and interactors they gleaned from DLG5 targeted proteomics. The quality of data and extent of investigation are truly remarkable. In the following I bring certain points to the authors' attention to raise this manuscript beyond reproach.

Again, we thank this reviewer for the positive evaluation of our manuscript and appreciating its significance for the community.

5) Clearly most the manipulations show drastic sarcomere assembly, fiber length and fiber formation phenotypes in DLM. Inferences from these phenotypes rely on the assumption that Hippo pathway proteins and others in this study are expressed in developing DLMs. Similar assumptions were made in Yatsenko et al. BMC Medicine https://doi.org/10.1186/s12916-019-1478-3 and Yadav, Puah and Wasser, Royal Society Open Biology 2016. This notion is mostly supported by consistent RNAi phenotypes.

We thank the reviewer for this comment. However, we do not only assume that Dlg5, Slmap and Hippo pathway components are expressed in developing DLMs. Our earlier systematic transcriptomics time-course using dissected flight muscles documented expression of all the components at mRNA levels (Spletter et al., 2018). To make this point clearer, we have now extracted the data concerning the genes of interest for this manuscript and display them again in new Figure 4—figure supplement 1C. Furthermore, we have raised antibodies against Dlg5 and Slmap protein and now document Dlg5, Slmap and Hippo protein expression and localisation in the developing DLMs in new Figure 2, Figure 2—figure supplement 1, Figure 4—figure supplement 1D, E.

6) To a skeptic, the drastic phenotypes seen may be an artefact of misexpression that may be irrelevant to native cellular processes i.e. these proteins are usually not expressed in DLMs but disturb development if introduced. Therefore, the epistasis analysis of these misexpression phenotypes and their transcriptional consequences, i.e. co misexpression experiments in Figure 1 to Figure 8, may not inform us of the native function of the proteins in DLM development. These results may reflect non-native protein function in a developmentally anachronistic tissue context. Heavy reliance on RNAi without ascertaining off-target affects in DLMs at the target stage of development (true for previous publications as well), further confounds inferences from these results. It may be essential to address such concerns. This point can simply be addressed in the Discussion.

To answer this sceptic comment, we now document endogenous mRNA and protein expression levels of most genes concerned (see comment above). Furthermore, we provide overwhelming genetic evidence for the phenotypes documented in our manuscript. Most genes were investigated with more than 1 RNAi line; together we provide data for 14 different RNAi lines, for 8 genes (*Dlg5, Slmap, Cka, Strip, yorkie, Hippo, warts*and *sd*) all of which are in the Hippo pathway. Furthermore, we provide gain of function phenotypes using variants of *UAS-Hippo*and *UAS-yorkie* constructs. In all cases the “Hippo loss of function” and “Hippo gain of function” phenotypes match perfectly. The specificity is further supported by the new data providing protein localisation and specificity of the antibodies as well as the documentation of the knock-down at protein level for Dlg5 and Slmap (new Figure 2—figure supplement 1).

7) The authors use mass spec investigations to elucidate protein-protein interactions to support their interpretations, specifically Dlg5GFP driven in muscles under the mef2-Gal4 promoter. While these interactions may happen in other muscles, data need to unequivocally show that they occur in developing DLMs at the stages being investigated. Since the source tissue for these experiments is whole pupae, and DLMs at these stages may not out-represent interactions in other muscles through this measurement, DLM specific comments invite skepticism. The Schnorrer lab has previously shown that molecular regulation of DLM growth differs from other *Drosophila* muscle. Therefore, examination of DLG5 protein-protein interactions in developing DLMs, on which subsequent connections with Hippo signaling have been based, would have been useful. Here again, no experiments are required, but this point can be addressed in the Discussion.

We indeed used 24-48 h APF pupae as samples for mass-spectrometry. However, at this developmental stage the flight muscles constitute most of the pupal muscle fibers, because larval muscle fibers have been dissolved and abdominal as well as leg myoblasts are still fusing and hence these muscles are small (Soler et al., 2004 Development, PMID 15537687; Weitkunat et al., 2017). Thus, most Dlg5-GFP was present in developing flight muscles. Proteomics of dissected developing flight muscles is not established and would have been very labour-intense. We added a comment in the Discussion to address this point.

8) Further, Dlg5 is a marker of T-Tubules (Peterson and Krasnow, 2015). Even if the proteomics were limited to DLMs, the authors would have to untangle the catastrophic effects of Dlg5 perturbation (Figure 1) in relation to Hippo signaling from oxygen availability to the developing muscle. The authors should address such a concern.

We have identified a role for Dlg5 in *Drosophila* muscles here for the first time. Peterson and Krasnow, 2015 have not used Dlg5, but Dlg1 as a marker for T-tubules. Neither did Peterson and Krasnow, 2015 manipulate Dlg1 genetically in this study. Hence, there is no link between Dlg5 and oxygen availability in muscle published to date.

Furthermore, manipulations of t-tubules have been done using an Amphiphysin mutant. However, this mutant results in normally sized flight muscles with normal myofibrils and no defect in muscle growth is reported (Razzaq et al. 2001).

Finally, blocking tracheal morphogenesis entirely in flight muscles by manipulating FGF secretion from flight muscles results in normally sized flight muscles as well (done independently in Peterson and Krasnow, 2015 and Sauerwald et al., 2019). Hence, tracheation is not required for flight muscle growth. In summary, we do not think the point of the reviewer is a valid concern that we need to address.

9) To ensure that their inferences are valid for DLMs specifically, the authors must first establish Hippo pathway protein expression and function. It is crucial that authors ascertain native expression of the Hippo, Yorkie and other member proteins of this signaling pathways specifically from isolated DLMs at the stages being investigated. Whole thoraces/pupae will not speak to DLM specific molecular processes. This can be achieved through IHCs. Visualizing Yki is notoriously difficult. Clean western blots from tagged transgenic lines showing normal development will suffice. The same procedures must be followed for known reporters of Yki function without overexpression. Further, if the expression of these proteins in the context of pupal DLMs can be demonstrated, the authors must show that these are in fact perturbed by manipulations. If these are not tenable experimentally, the authors should discuss the issue of the phenotypes upon driving expression, using the mef2-Gal4 driver, of Yki, YkiCA, Yki-myr, Hippo and Diap1, but not Dlg5; when we have no evidence for their expression in the muscles. Could this be an ectopic gain-of-function phenotype? If the authors can show expression data if they have them that will be nice, or else they should ratchet-down their interpretation appropriately. (The authors could show native expression and activity of Yki in developing DLMs, through IHC or WB through tagged Yki expression if necessary. Reporters like diap1-GFP and ex-LacZ among others are also useful. Western blots or mass spectrometry from isolated developing DLMs of Yki, p-Yki and other Hippo pathway members are acceptable as well. If the effects of genetic manipulations on the native levels and function of these proteins can be shown, that will be very good. Else, please discuss potential concerns).

Prior to this study, targets of Yorkie in the muscle were unknown. Hence, instead of using reporter lines that were established in epithelial cells we decided to identify Yorkie targets in flight muscle systematically on RNA level using dissected isolated DLMs at the important stages and the various gain and loss of genotypes (see Figure 8). These experiments show that neither *diap1* nor *ex* are prime targets of the Hippo pathway in muscle and instead we have identified novel regulated targets in flight muscles.

We have confirmed the perturbed expression at the protein level for 2 identified targets (Act88F and Mhc, see Figure 8). In the revised version we have added additional quantitative data that document changes for 2 more sarcomeric protein components (Prm and Sls) in gain and loss of function Yki conditions (new Figure 9—figure supplement 1).

Furthermore, we now do provide protein expression and localisation data for Hippo, Dlg5 and Slmap in the DLMs throughout development (new Figure 2 and Figure 4—figure supplement 1). Together, this documents the localisation and function of Hippo pathway components in flight muscles.